# A structural vista of phosducin-like PhLP2A-chaperonin TRiC cooperation during the ATP-driven folding cycle

Junsun Park [1,4], Hyunmin Kim[1,4], Daniel Gestaut[2,4], Seyeon Lim[1], Kwadwo A. Opoku-Nsiah [2], Alexander Leitner [3], Judith Frydman [2,5] ✉ & Soung-Hun Roh [1,5] ✉

Proper cellular proteostasis, essential for viability, requires a network of chaperones and cochaperones. ATP-dependent chaperonin TRiC/CCT partners with cochaperones prefoldin (PFD) and phosducin-like proteins (PhLPs) to facilitate folding of essential eukaryotic proteins. Using cryoEM and biochemical analyses, we determine the ATP-driven cycle of TRiC-PFD-PhLP2A interaction. PhLP2A binds to open apo-TRiC through polyvalent domain-specific contacts with its chamber's equatorial and apical regions. PhLP2A N-terminal H3-domain binding to subunits CCT3/4 apical domains displace PFD from TRiC. ATP-induced TRiC closure rearranges the contacts of PhLP2A domains within the closed chamber. In the presence of substrate, actin and PhLP2A segregate into opposing chambers, each binding to positively charged inner surface residues from CCT1/3/6/8. Notably, actin induces a conformational change in PhLP2A, causing its N-terminal helices to extend across the inter-ring interface to directly contact a hydrophobic groove in actin. Our findings reveal an ATP-driven PhLP2A structural rearrangement cycle within the TRiC chamber to facilitate folding.

Proper protein folding and homeostasis (proteostasis), fundamental to maintaining cellular integrity, depend on the action of molecular chaperones. While in vitro, many molecular chaperones successfully promote the folding of non-native polypeptides, in the cell, efficient proteostasis involves the cooperation of distinct chaperones. The specificity and structural basis for such cooperation is poorly understood for most chaperone systems, and in particular for the eukaryotic chaperonin TRiC/CCT.

The ring-shaped chaperonin TCP-1 ring complex (TRiC, also called CCT) is a ~1 MDa hetero-oligomeric complex that has a double-chamber architecture, with each octameric ring formed by subunits CCT1-8[1–3]. TRiC-mediated protein folding depends on an ATP-driven conformational cycle, which regulates the opening and closing of a built-in lid over

the central chamber of each ring[4,5]. In the apo-state, the lid is open and substrates bind to the apical domains of specific CCT subunits[6,7]. ATP hydrolysis leads to lid formation and substrate encapsulation in the closed central chamber, where folding occurs[6,8–10]. Previous biophysical studies suggested stepwise folding within the chamber[6,11] and recent cryoEM analysis revealed that TRiC mediates a domain-wise assembly of the substrate through a directed folding pathway[12]. TRiC plays a unique role in cellular proteostasis, as it is obligately required to facilitate the folding of ~10% of the eukaryotic proteome, including many essential proteins with a complex topology that cannot fold spontaneously or with the aid of any other chaperone[6,13–15].

In the cell, TRiC functions in cooperation with many cochaperones, including Prefoldin (PFD, also called GIMc) and phosducin-like

[1]School of Biological Sciences, Institute of Molecular Biology and Genetics, Seoul National University, Seoul, South Korea. [2]Dept of Biology and Genetics, Stanford University, Stanford, CA 94305, USA. [3]Institute of Molecular Systems Biology, Dept of Biology, ETH Zurich, Zurich 8093, Switzerland. [4]These authors contributed equally: Junsun Park, Hyunmin Kim, Daniel Gestaut. [5]These authors jointly supervised this work: Judith Frydman, Soung-Hun Roh. ✉e-mail: jfrydman@stanford.edu; shroh@snu.ac.kr

proteins (PhLPs)[16]. Prefoldin is a jellyfish-shaped hetero-oligomeric complex that binds the open state of TRiC through specific contacts with the apical domains but is released upon ATP-driven lid closure[17]. PFD maintains substrates in a dynamic unstructured conformation[12] and enhances the processivity of TRiC-mediated folding by transferring substrates to the open TRiC cavity[17]. The PhLPs are a family of ~30 kDa cytosolic proteins that have also been shown to regulate TRiC-mediated protein folding[18]. There are five distinct PhLP homologs in humans and two homologs in yeast[19–21]. All PhLP homologs share a similar domain organization, with a central thioredoxin-like domain[22] flanked by variable length flexible N- and C-terminal domains[19]. While the precise function and mechanism of PhLPs in TRiC-mediated folding are not very clearly understood, each PhLP isoform is reported to have distinct activity and specificity for different TRiC substrates. For instance, PhLP1 assists TRiC-mediated folding and heterodimer assembly of Gβγ and Gβ₅-Regulators of G protein signaling (RGS)[23,24]. In vitro experiments have shown that human PhLP2A, PhLP2B, and PhLP3 can impact TRiC-mediated folding of actin and tubulin[25,26] and yeast PLP2, a homolog of PhLP2A stimulates the actin folding[25]. While yeast PLP1 is dispensable for viability, yeast PLP2 is essential and interacts genetically with both tubulin and PFD-subunit Pac10[20,27] Of note, PhLP2A has been mapped as a genetic determinant of a myopathy related disease[28].

Previous structural analyses of TRiC reported a low-resolution EM structure of PhLP bound to open TRIC[29] and more recently cryoEM structures of an actin-PhLP2A-TRiC complex that presumably represents a late stage intermediate after ATP hydrolysis[30]. However, how PhLPs engage TRiC throughout the ATP-driven conformational cycle remains poorly defined. In this study, we used purified components to reconstitute actin folding by TRiC with its cochaperones PFD and the ubiquitous PhLP isotype PhLP2A to understand the structural and mechanistic interplay between PhLP2A and TRiC throughout the ATP-driven chaperonin cycle. Thereby we report a structural and biochemical analysis of the complex of TRiC and PhLP2A in distinct ATP hydrolysis states. Our study provides structural and functional insights into how PhLP2A assists TRiC-mediated substrate folding.

## Results

### The architecture of PhLP2A complex with open apo-state TRiC
The complex between open human TRiC and PhLP2A was generated using purified components and subjected to cryoEM to investigate its structure. We obtained a 3.08 Å resolution consensus map (Fig. 1a, Supplementary Fig. 1a–d) containing PhLP2A-dependent density within the TRiC chamber, positioned close to the equatorial region. The overall TRiC architecture closely mirrors previously established structural characteristics[17,31], maintaining robust equatorial domains and dynamic apical regions. Additionally, we noted a consistent pattern of partial nucleotide occupancy, with a more prominent nucleotide density observed in the lower affinity hemisphere of TRiC[9], particularly in CCT6, CCT8, and CCT3, in contrast to the higher affinity side of TRiC, represented by CCT2, CCT5, and CCT7 (Supplementary Fig. 2). Subsequent 3D classification and refinement revealed high-resolution features for this extra density (Supplementary Fig. 1b). We then fitted the crystal structure from the thioredoxin-fold domain of human PhLP2B to the density (PDB code: 3EVI)[22] and the model aligned well with many bulky side chains corresponding to the cryoEM density. This provides strong evidence that the extra density represents PhLP2A (Fig. 1b).

PhLP2A consists of a coiled-coil N-terminal domain (amino acids (aa) 1–90, NTD) followed by a thioredoxin-body domain (aa 91–210, TXD) and a short C-terminal domain (aa 211–239, CTD) (Fig. 1e-(i)). The TXD and CTD density allowed the tracing of the backbone based on sidechain densities. However, we could not detect any density attributable to the NTD. We, therefore, performed focused 3D classification on each CCT subunit (Supplementary Fig. 1b). This analysis

resolved an extended helical density that can be attributed to helix 3 (H3) of the NTD of PhLP2A adopting two different orientations associated with either the apical domain of CCT3 or CCT4 (Fig. 1c, d). Next, we performed AlphaFold prediction[32] for each possible PhLP2A-CCT subunit pair to predict residues contributing to the potential interaction. Notably, only the apical domains of CCT3 and CCT4 were predicted to form a dimeric complex with H3 of PhLP2A with good prediction scores (Supplementary Fig. 3a, b), in agreement with our cryoEM density maps. Therefore, we built atomic models of the complex between open TRiC and residues 63–232 of PhLP2A (Fig. 1c, d(i)), based on experimental densities as well as predicted CCT3/4 and PhLP2A contacts (Fig. 1d(ii, iii), Supplementary Fig. 3a, b).

To independently examine the architecture of the TRiC and PhLP2A complex, we next performed crosslinking mass spectrometry (XL-MS) analyses. The intra- and inter-molecular crosslinks observed supported and extended our cryoEM-derived model. While H1/H2 of PhLP2A was not identified in the cryoEM map, XL-MS identified multiple crosslinks between H1/H2 and the other PhLP2A domains. Mapping these crosslinks onto a predicted model of full-length PhLP2A (Fig. 1e, f) suggests the NTD bends to place H1/H2 in the proximity of TXD. In good agreement with our PhLP2A-TRiC, we detected many intermolecular crosslinks between PhLP2A H3 and CCT3-CCT4 as well as between the PhLP2A TXD and CTD domains to the equatorial domains of CCT3 and CCT6 respectively (Fig. 1g(i, ii)). Notably, we detected multiple crosslinks between the unresolved PhLP2A helices H1 and H2 with the cavity facing residues of CCT1, and CCT4. These XL-MS results together with the cryoEM structure are consistent with the PhLP2A NTD localizing inside the open TRiC chamber (Fig. 1h).

### Molecular contacts between TRiC and PhLP2A in the open conformation
The cryoEM-derived model revealed that domain-specific molecular contacts mediate the PhLP2A-open TRiC interaction (Fig. 2a–c and Supplementary Movie 1). Overall, PhLP2A binds within the open TRiC chamber in an extended conformation where each of its domains engages distinct subunit-specific sites within the chaperonin. The central PhLP2A TXD domain is encapsulated in the open TRIC chamber where it is constrained through polar and hydrophobic interactions with the equatorial domains of CCT3 and CCT1 (Fig. 2a(iii), b, c, and Supplementary Movie 1). The CTD engages in an interface formed by CCT3 and CCT6 through a single amphipathic helical structure with a highly hydrophobic side (Fig. 2a(iv, v)). This hydrophobic side of the CTD helix forms a hydrophobic zipper motif with an equatorial domain helix of CCT6 at the interface with CCT3 (Fig. 2a(iv, v) and Supplementary Movie 1). The NTD of PhLP2A can adopt two different orientations, whereby H3 of the NTD contacts the apical regions of either CCT3 or CCT4. While the NTD is mostly negatively charged, H3 has a unique positively charged patch (residue 75–78), which displays a high degree of complementary to the negatively charged apical domains of CCT3 and CCT4 (Fig. 2a(i, ii), b). While we could not resolve H1 and H2 of the NTD, we observe XL-MS contacts between these elements and the TRiC chamber including the equator of CCT1, intermediate domains of CCT3/4 and apical domains of CCT2/6. Of note, the intermediate domains of CCT3 and CCT4 have wide surfaces of positively charged residues, which may interact with the negatively charged H1 and H2 of the PhLP2A NTD (Figs. 1g, h and 2b).

### PhLP2A modulates the PFD-TRiC interaction
Next, we examined the interplay between PhLP2A and cochaperone PFD, which also interacts with the open state of TRiC through the apical domains of CCT3 and CCT4 (Supplementary Fig. 4a). Comparing the intermolecular interfaces of CCT3/4 with PhLP2A or PFD (PDB:7WU7) showed both TRiC interactors engage in similar salt bridges with the same CCT subunits (Fig. 3a). When the two models are superimposed, the contact sites of PFD and PhLP2A with CCT3 and

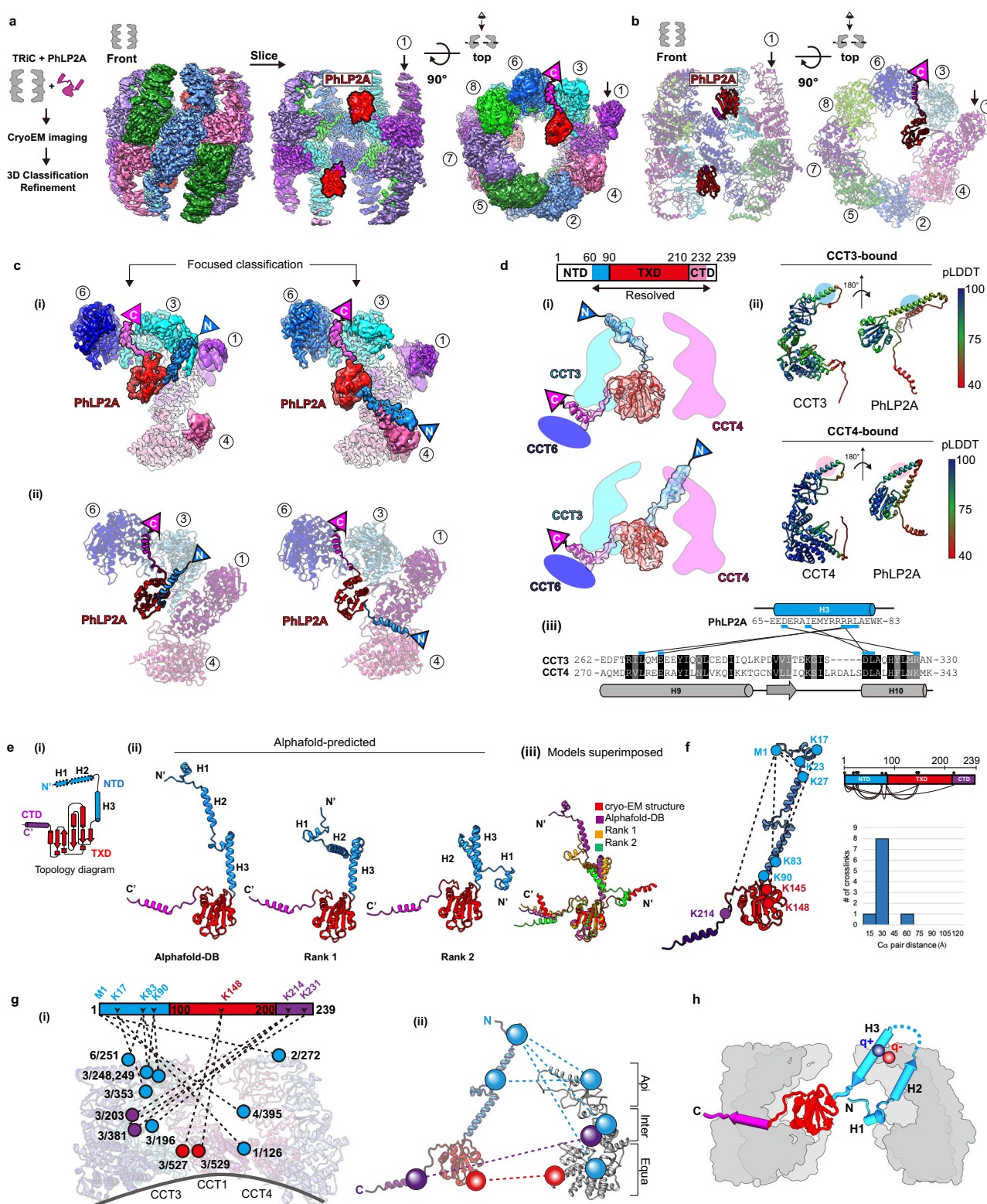

CCT4 significantly overlap each other (Fig. 3a, Supplementary Fig. 4a(ii)). We hypothesized the steric clash between PFD and PhLP2A may lead to mutually exclusive binding, thus creating a competitive relationship between PFD and PhLP2A for TRiC binding.

Given the partial overlap of PhLP2A and PFD bindings sites on open TRiC we examined our ability to generate a ternary complex with both cochaperones. To generate complexes of PhLP2A and PFD with open TRiC we used a TRiC:PFD:PhLP2A ratio of 1:2:~1, i.e. a two-fold ratio of PFD to TRiC and PhLP2A. When analyzed by cryoEM, the

particles displayed high heterogeneity with respect to the binding states of the cochaperones to TRiC (Supplementary Fig. 4b, c). We performed 3D-focused classification on each location of PFD and PhLP2A independently and found that TRiC particles only contained either PFD or PhLP2A (Fig. 3b, Supplementary Fig. 4c). This observation indicates PFD and PhLP2A binding to TRiC is mutually exclusive.

We next explored the interplay of these cochaperones on binding open TRiC. We used a native gel-based mobility shift to determine the apparent $K_d$ of PhLP2A for open TRiC to be 170 nM (Fig. 3c). Previous

**Fig. 1 | CryoEM structure of the PhLP2A-open TRiC complex. a** (Left) Schematic of the designed experiment. (Right) Front, slice, and top view of consensus map of PhLP2A encapsulated in the TRiC folding chamber. Each CCT subunit and PhLP2A are color-coded as defined in the top view. **b** Atomic model of the PhLP2A-TRiC complex from the front and top view. CCT subunits and PhLP2A are color-coded as in **a. c** Two binding modes of the PhLP2A NTD are revealed by 3D-focused classification. (i) Densities of CCT3 or CCT4 bound to PhLP2A. (ii) Atomic models of the complexes of the PhLP2A NTD and CCT3 or CCT4 using AlphaFold prediction. Each domain of PhLP2A is color-coded: NTD: dodger blue, TXD: red, CTD: dark magenta. **d** (i) Schematic diagram of PhLP2A and the CCT3/4 complex. The colored box indicates the modeled residues of PhLP2A with each domain color-coded. (ii) AlphaFold prediction of each complex with a per-residue confidence score (pLDDT) diagram. (iii) Sequence alignment of H9 and H10 of CCT3 and CCT4 at the major interaction site of PhLP2A. Predicted interactions between CCTs and PhLP2A are labeled with black lines. **e** (i) 2D topology diagram of PhLP2A. Each domain is color-coded as indicated in **c.** (ii) Three representative structures of AlphaFold-predicted full-length models of PhLP2A, color-coded in a domain-wise manner and (iii) superposition of the three models with the experimental model. **f** Intramolecular crosslinking of PhLP2A. (Left) Intramolecular crosslinks labeled on the PhLP2A model. (Top, right) 2D schematic diagram of intramolecular crosslinks on PhLP2A. N-terminal helices which were unresolved in cryoEM structures crosslink to all the other domains. (Bottom right) Graph of Cα pair distance. **g** Intermolecular crosslinking between PhLP2A and CCT subunits. (i) Detailed representation of the crosslinks on the model. PhLP2A crosslinks to the residues on CCT subunits of the half hemisphere of CCT2/4/1/3/6. (ii) Schematic diagram showing intermolecular crosslinks between each domain of the PhLP2A and a CCT subunit. **h** Schematic representation of PhLP2A topology in an open TRiC chamber. While CTD and TXD are anchored near the equatorial domain of CCT3/6, N-terminal helices can adopt various topologies and reside inside the folding chamber.

analyses using this assay showed the apparent $K_d$ of PFD for open (apo) TRiC is remarkably similar at $K_d$ = ~160 nM[17]. Subsequently, we carried out reciprocal competition experiments to evaluate the ability of either PFD or PhLP2A to displace the other cochaperone from a pre-formed complex with open TRiC (Fig. 3d). Briefly, fluorescently labeled cochaperones were bound to GFP-TRiC and their displacement from the TRiC complex by unlabeled cochaperone was determined using a native gel mobility shift assay. Strikingly, despite their similar overall apparent $K_d$ values, we observe an unequal ability of PFD and PhLP2A to displace the other cochaperone. While PhLP2A displaced up to 70% of TRiC-bound PFD, similar concentrations of PFD resulted in virtually no displacement (<10%) of TRiC-bound PhLP2A. We rationalize these observations based on the distinct mode of binding of these TRiC cochaperones. While PFD binds primarily to a single site in the apical domain of CCT4, PhLP2A binds in a multivalent manner, with binding sites in both the apical domains of CCT3/4 (which overlaps with that of PFD) and the equatorial domain regions of CCT3/6. These additional interactions potentially contribute to PhLP2A's enduring association with TRiC even when PFD is added in excess. Thus, taken together with our structural data that PFD and PhLP2A both engage with TRiC through common binding interfaces, these experiments suggest a hierarchical association of these cochaperones with TRiC.

Our analyses showing both PFD and PhLP2A associate with TRiC through an electrostatic interface with subunits CCT3/CCT4 suggest a mechanistic underpinning to orchestrate their action on a TRiC-centered network. To assess the functional relevance, we asked if these interaction interfaces are conserved in evolution. Indeed, the charged patch on CCT4 is highly conserved and its mutation impairs cellular proteostasis[17]. Residue conservation analysis for each chaperone[33] indicates that residues forming the contact surface among TRiC, PFD and PhLP2A are evolutionary conserved. Particularly, PFD6 has a highly conserved positively charged surface that contacts CCT4 (Supplementary Fig. 4f). The charged H3 residues in PhLP2A that contact CCT4 are also highly conserved (Supplementary Fig. 4f). Finally, the contact areas of CCT3 and CCT4 toward PFD and PhLP2A are evolutionarily conserved (Supplementary Fig. 4g), as described. The evolutionary conservation of the inter-chaperone interaction sites suggests a co-evolution between these three chaperones to coordinate substrate delivery and TRiC-mediated folding.

### The architecture of PhLP2A complex with ATP-AlFx induced closed state TRiC

TRiC conformation changes upon ATP hydrolysis, which induces lid closure over the central chamber[5,8,34]. Since lid closure occludes the PFD binding site in CCT3/4 and induces the release of bound PFD from TRiC, we examined if PhLP2A forms a complex with closed TRiC. In a prior study, Jin et al. demonstrated that TRiC maintains an identical structure with full nucleotide occupancy in the ATP binding pockets under both ATP and ATP/AlFx conditions[4]. Since AlFx captures the

TRiC complex in a stable post-hydrolyzed state during TRiC's ATP hydrolysis cycle[12,30,35,36], we chose to employ the nucleotide analog ATP/AlFx to attain high-resolution structures. We incubated the TRiC: PhLP2A in a 1:4 molar ratio with ATP/AlFx, which generates a stably closed TRiC conformation and subjected it to cryoEM analysis (Fig. 4a(i)). The consensus map of closed TRiC reached 2.95 Å resolution with significant additional density inside the chamber. We validated that our closed TRiC structure resembled that of the native ATP-induced structure including full nucleotide occupancy in a manner similar to what is observed in ATP-induced structures[4,36] (Supplementary Fig. 6). Subsequently, focused classification on the inner TRiC chambers identified subpopulations of PhLP2A in either one (14.0%) or both chambers (9.5%) (Supplementary Fig. 5a–d). The TRiC sub-population with PhLP2A in both chambers was used to further refine the final map of the PhLP2A protein fully encapsulated inside the closed TRiC chamber at 3.24 Å resolution (Fig. 4a(ii, iii)). Overall, side chain details for TRiC were clearly revealed on the map allowing us to build an atomic model. For PhLP2A, we could build a refined model spanning residues 27–214 (Fig. 4b(i), Supplementary Fig. 5e). PhLP2A H1-2 (aa1–26) in the NTD and the CTD (aa 215–239) are not well resolved but have attributable densities that can be observed at a lower contour providing an approximate location of the NTD in proximity to CCT1 and CCT3 and of the CTD in proximity to the intermediate domains between CCT1 and CCT4 (Fig. 4b(ii)).

Remarkably, PhLP2A also interacts with the closed TRiC chamber through domain-specific contacts with specific CCT elements, albeit these are distinct from those observed in the open state. Negatively charged residues in the NTD form multiple salt bridges with positively charged residues exposed by CCT3 and CCT6 in the closed chamber, these contacts constrain the NTD to bind this region of the TRiC inner wall (Fig. 4c(i), Supplementary Movie 2). The TXD localizes beneath the lid region through polar and hydrophobic interactions with the lid regions of CCT5, 2, and 4 (Fig. 4c(ii), Supplementary Movie 2). The CTD density at the lower contour is in proximity to the interface between the intermediate domains of CCT1/4. Previous analysis recognized that the TRiC inner chamber contains an asymmetric charge distribution, with a strong positively charged inner surface contributed by subunits CCT1/3/6/8 and a strong negatively charged inner surface contributed by CCT7/5/2/4[1] (Fig. 4d). The strong electrostatic binding between the negatively charged NTD of PhLP2A and the positive TRiC hemisphere together with the interaction between TXD of PhLP2A and the TRiC lid segments on the other hemisphere (Fig. 4d, e) establishes a diagonal binding topology of PhLP2A inside the closed TRiC chamber (Fig. 4c, Supplementary Movie 2).

### ATP-driven TRiC closure changes the conformation of bound PhLP2A

Our analysis shows that, unlike PFD, PhLP2A binds TRiC in both open and closed states. ATP hydrolysis leads to many changes in available

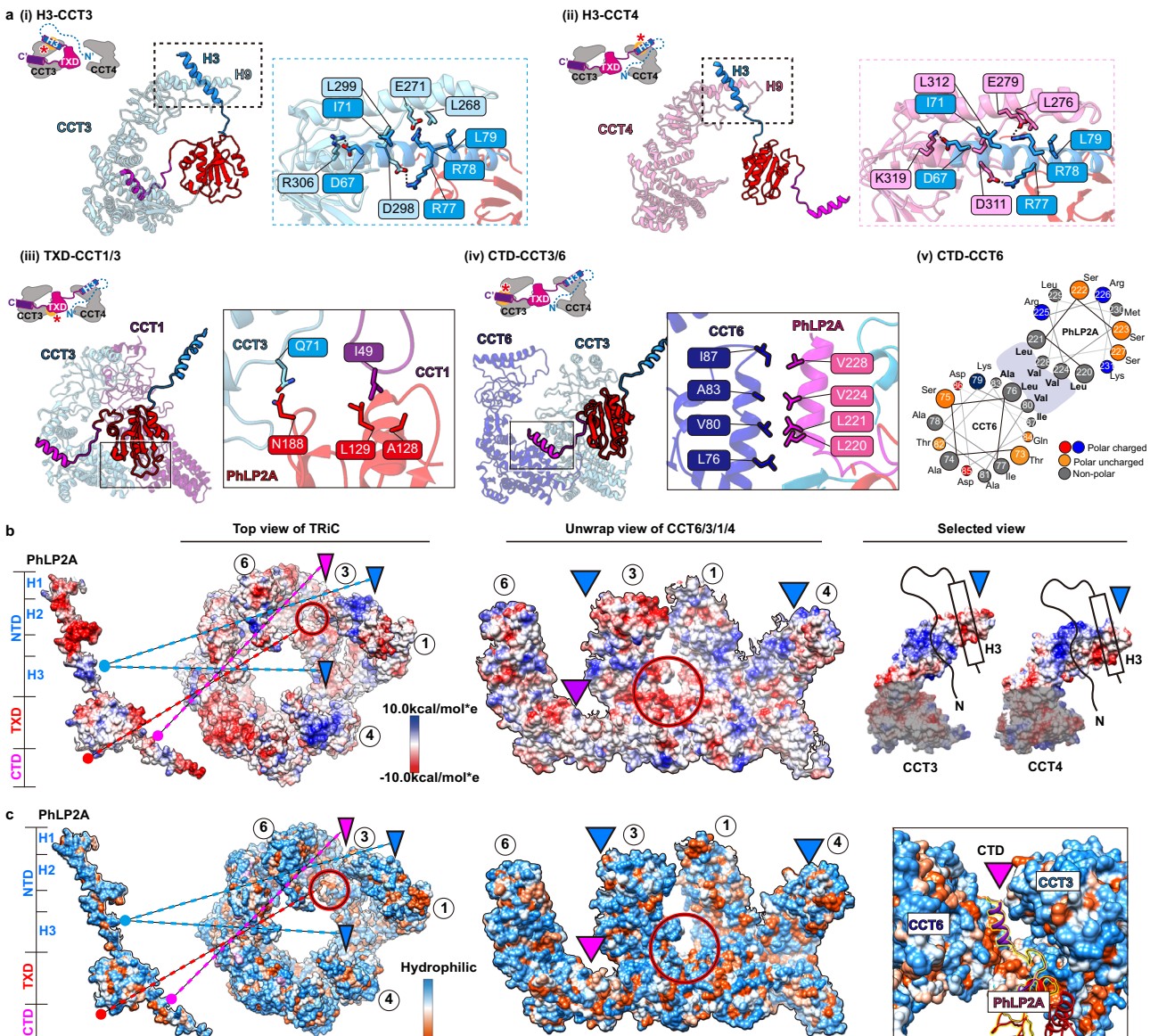

**Fig. 2 | Molecular contacts between PhLP2A and open TRiC chamber. a** Detailed interaction between PhLP2A and TRiC. (i-ii) Zoomed-in view of the binding between PhLP2A H3 and the apical domain of CCT3 or CCT4 based on AlphaFold prediction. H9 and H10 of each CCT subunit are major interaction sites; their binding modes are similar either on the topological or sequence level. (iii) Interaction between the lower part of TXD and the nucleotide-sensing loop of CCT1. (iv) Helix-to-helix interaction between PhLP2A CTD and CCT6. (v) Helix wheel diagram on PhLP2A CTD-CCT6 showing the amphipathic nature of the helix. **b** Electrostatic surface charge of PhLP2A and open TRiC chamber. PhLP2A, the top view of TRiC, and unwrap view of the CCT4/1/3/6 half-hemisphere are displayed. Binding sites between PhLP2A and TRiC are indicated with color-coded lines and triangles (dodger blue for the NTD and dark magenta for CTD of PhLP2A). The red circle indicates the TXD binding site to TRiC. Selected views focus on the part of the apical domain where H3 of PhLP2A binds via electrostatic charge interaction. **c** Surface hydrophobicity of PhLP2A and the open TRiC chamber. Interactions between PhLP2A and TRiC are as indicated in **b**. While other binding sites are mainly hydrophilic, the TXD binding interface between CCT1/3 and the CTD binding crevasse between CTD3/6 are hydrophobic.

TRiC interaction interfaces, including in the apical domains, the formation of an asymmetrically charged inner chamber wall, and the formation of the lid all of which produce a global reorientation of the PhLP2A domains within the TRiC chamber (Fig. 5a, b, Supplementary Movie 3). The transition from open to closed TRiC triggers a ~50 Å movement of the negatively charged NTD H3 of PhLP2A from binding the apical domains of CCT3 and CCT4 to binding the positive inner wall of the closed TRiC at CCT3 and CCT6 intermediate domains (Fig. 5a, b, Supplementary Movie 3). In addition, TXD rotates ~180 degrees and moves ~30 Å upon lid closure, from equatorial contacts to CCT3-CCT6 in the open state to contacting the opposite hemisphere at the apical CCT5/2/4 closed lid segments (Fig. 5a, b). Finally, the PhLP2A CTD, which is also tightly anchored to the CCT3/6 interface in open TRiC, is

released in the folding chamber upon TRiC closure, with weak contact with the CCT1/4 inner wall. In conclusion, the changes in surface residues within the TRiC chamber upon lid closure induce a dramatic conformational repositioning of PhLP2A by reassortment of its domain interactions with the chamber.

## PhLP2A domain-specific interactions with open and closed TRiC
To better define the interactions of PhLP2A domains with specific TRiC subunits, we generated four domain variants: NTD (aa 1–84), TXD (aa 85–211), NTD-TXD (aa 1–211) and TXD-CTD (aa 85–239) respectively (Fig. 5c(i)). To assess TRiC binding, we incubated either full-length PhLP2A or four domain variants with bovine TRiC (0.5 µM TRiC, 5 µM PhLP2A variant) (Fig. 5c(ii), Supplementary Fig. 7a). Binding was

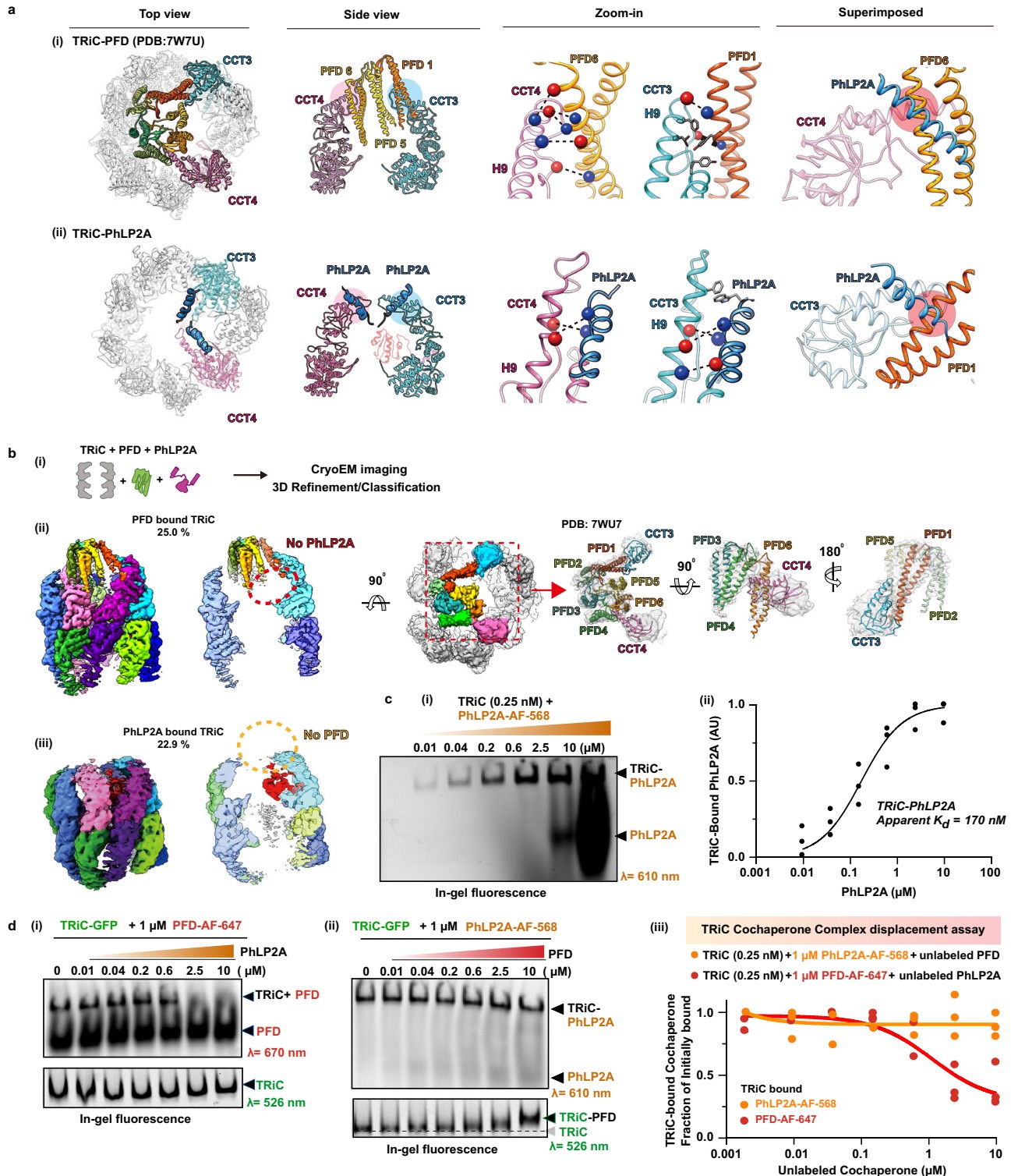

**Fig. 3 | PhLP2A modulation on the PFD-TRiC network. a** Atomic models of (i) PFD-bound TRiC (PDB:7WU7) and (ii) PhLP2A-bound TRiC. CCT3, CCT4, PFD and the PhLP2A NTD are highlighted. Contacts between TRiC and PFD or PhLP2A are indicated as colored-circles in the side view. Zoomed-in view of each contact between TRiC (CCT3, CCT4) and the cochaperones (PFD, PhLP2A). Negatively charged residues are indicated by red balls while positively charged residues are indicated by blue balls. Other residues are shown as stick cartoons. PFD and PhLP2A models are superimposed on their binding sites of CCT3 or CCT4. Red circles indicate the clash between the molecules. **b** (i) Schematic of the designed experiment. (ii, iii) 3D classification reveals PFD-bound TRiC (25.0%) and PhLP2A-bound TRiC (22.9%), respectively. The PFD-TRiC atomic model (PDB: 7WU7) fitted to the PFD-bound TRiC density. The red circle indicates no detectable PhLP2A density while the yellow circle indicates no detectable PFD density. **c** (i) Representative immunoblot of serially diluted PhLP2A (0.01–10 μM) bound to TRiC (0.25 μM). (ii) Quantification of immunoblots of dose dependent TRiC-bound PhLP2A establishes apparent dissociation constant. **d** (i) Native gel analysis using GFP-TRiC and cochaperones. Serially diluted PhLP2A (0.01–10 μM) were applied to the mixture of GFP-TRiC (0.25 μM) and AF-647 labeled PFD (1 μM). (ii) Serially diluted PFD (0.01–10 μM) were applied to the mixture of GFP-TRiC (0.25 μM) and AF-568 labeled PhLP2A (1 μM). (iii) Quantification of PhLP2A (orange) displaced from TRiC by PFD binding, and PFD (red) released as a result of PhLP2A binding. Source data are provided as a Source Data file.

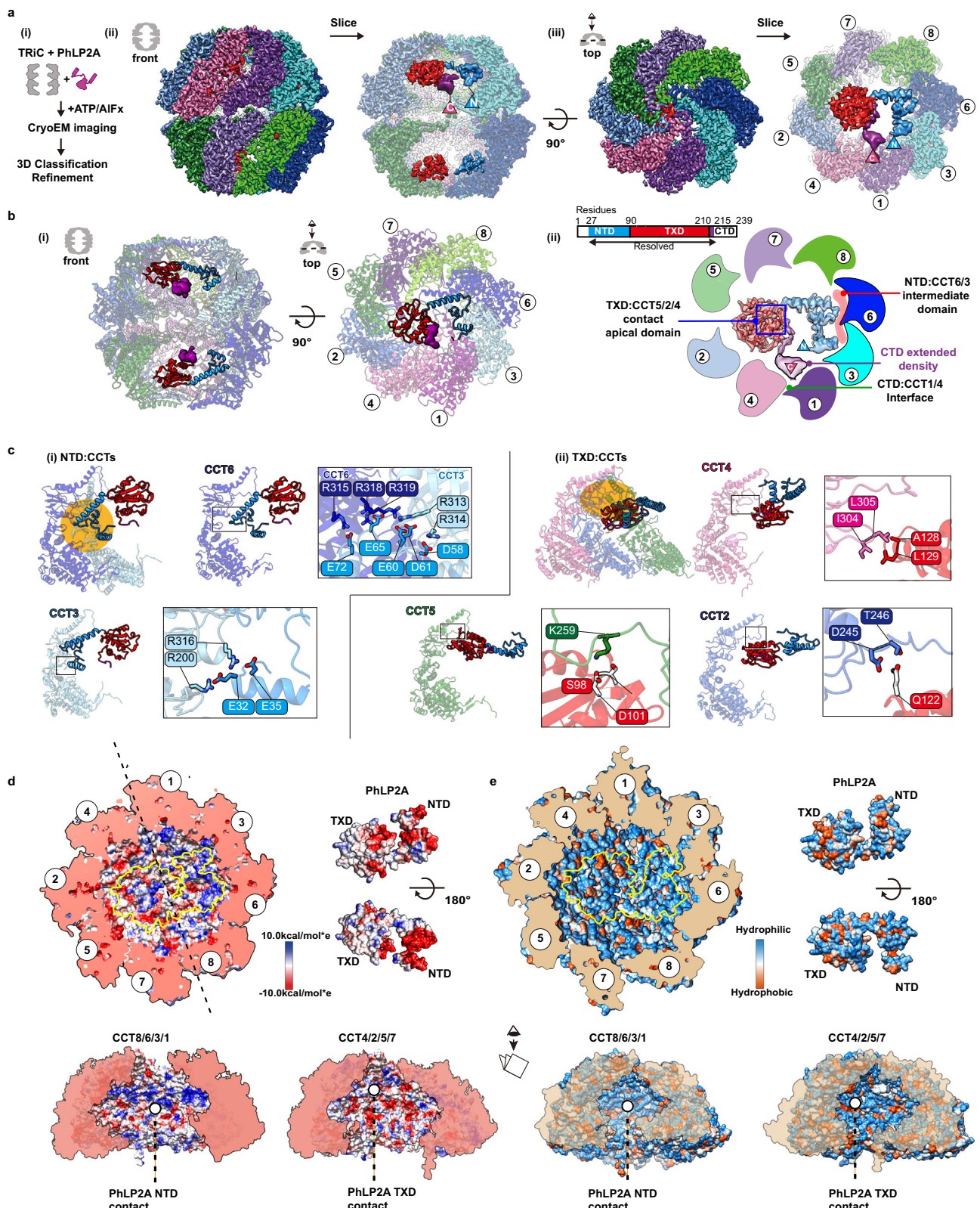

assessed by N-PAGE followed by immunoblot against the His-tag on PhLP2A to examine if the PhLP2A variants comigrated with either open or ATP-closed bovine TRiC (Fig. 5c(ii), Supplementary Fig. 7a). While the NTD alone does not bind TRiC, the TXD alone bound TRiC in both the absence or presence of ATP/AlFx. Of note, the TXD-CTD variant increased TRiC binding over TXD alone. We conclude that the TXD is sufficient to mediate TRiC-binding but the CTD enhances this interaction. Of note, these domain variants can still be encapsulated by TRiC lid closure (Fig. 5c, Supplementary Fig. 7a).

To further dissect the molecular determinants of TRiC binding, complexes of PhLP2A domain variants with open and closed TRiC were examined by cryoEM (Fig. 5d, Supplementary Fig. 7b, c). For the open TRiC complexes, we identified extra density attributable to the PhLP2A variants with similar placements as identified in full-length PhLP2A.

**Fig. 4 | Structure of PhLP2A inside TRiC chamber after chamber closing.**
**a** CryoEM structure of PhLP2A inside the closed TRiC chamber after ATP hydrolysis. (i) CryoEM imaging preparation scheme. (ii) Front and slice view of the closed PhLP2A-TRiC complex. N- and C-terminus of PhLP2A are labeled in the map. (iii) Top and slice view of the closed PhLP2A-TRiC complex. Due to the varying resolution, the PhLP2A density is shown at different thresholds according to the protein domains. Note that both of the folding chambers can be occupied by PhLP2A, as in open-state TRiC. Each subunit and domain are color-coded as in Fig. 1. **b** (i) Model of PhLP2A inside the closed TRiC chamber. The CTD is not modeled and the lowpass filtered electron density is shown instead. (ii) Summary of binding sites between PhLP2A and the closed TRiC chamber. Compacted NTD helices interact with the intermediate domain of CCT3/6 (colored in red) and the following TXD interacts with the apical domain of CCT5/2/4 (marked by a blue box). Note that although

these domains are not resolved to the atomic level, the density map at the lower contour level shows the NTD close to CCT1/3 and the CTD close to the equatorial domains between CCT4/1. **c** Zoom-in view of the domain-wise interaction between PhLP2A and CCT subunits. (i) Molecular contacts between CCT 3/6 and the NTD of PhLP2A and (ii) between CCT5/2/4 and the TXD of PhLP2A. Contact areas are indicated by a yellow circle. Residues making contacts are displayed and labeled. **d** Electrostatic surface charge showing charge complementarity between PhLP2A and the closed folding chamber. The positively charged half hemisphere of CCT1/3/6/8 provides a binding surface for the negatively charged PhLP2A NTD.
**e** Hydrophobic surface charge of closed folding chamber and PhLP2A. While the inner wall of the chamber is mostly hydrophilic and charged, part of the surface composed of CCT5/2/4 provides hydrophobic patches for PhLP2A TXD to bind on.

The NTD-TXD fragment in the TRiC open chamber showed similar density attributable to the NTD around the apical helices of CCT3/CCT4 but displayed higher heterogeneity around the TXD density (Fig. 5d(i)). TXD-only showed density close to the TRiC equator only at a lower contour level, suggesting its intrinsic affinity for the TRiC chamber depends on other PhLP2A domains (Fig. 5d(ii)). Indeed, TXD-CTD yielded high-resolution features like WT PhLP2A, with the CTD anchored between CCT3/CCT6 and the TXD close to CCT3 (Fig. 5d(iii)). The complex particle population of TXD-CTD was notably higher and the structural features were sharper compared to NTD-TXD and TXD only, indicating that CTD plays a key role in PhLP2A binding to open TRiC, consistent with N-PAGE (Fig. 5c, d, Supplementary Fig. 7c). The addition of ATP/AlFx encapsulated all PhLP2A variants and rearranged their interactions within the closed TRiC chamber (Fig. 5c). While NTD-TXD reoriented similar to full-length PhLP2A in the closed TRiC chamber, we could not observe any density attributable to either TXD or TXD-CTD in closed TRiC (Fig. 5d), even though N-PAGE indicated all PhLP2A variants remain associated with closed TRiC. This indicates that TXD and TXD-CTD are encapsulated upon closure but interact weakly and dynamically with the closed chamber (Fig. 5d(ii, iii)).

This structural and biochemical analysis reveals the complex role of PhPL2A domains in the interaction with TRiC throughout its ATP-driven conformational cycle. In the open state, it appears TXD and the CTD play key roles in binding and encapsulation. On the other hand, the NTD is an essential component for the proper structural orientation of PhLP2A in the closed TRIC chamber. Our analysis also provides insight into the domain binding and release events leading to PhLP2A rearrangements during the chamber closing process. Each TRiC ring is composed of eight distinct subunits with distinct ATP binding affinities[9,37,38]. As a result, an asymmetric wave of chamber closing has been suggested[4,9]. Upon closure, each CCT subunit undergoes conformational changes induced by ATP hydrolysis within an allosteric network communicating equatorial and apical domains (Fig. 5e(i, ii)). We speculate that lid closure releases the NTD from its charged binding sites at the apical domains of CCT3/CCT4, and instead anchors it to the charged surface of the closed chamber at CCT3/6 (Fig. 5e(ii, iii)). This rearranges and orients PhLP2A within the closed folding chamber. The closure also weakens the open-state interactions of the CTD with CCT3/6, and of the TXD with CCT3 allowing them to be dynamically mobile within the closed chamber. The anchoring effect of the NTD in the closed state may enhance the affinity of TXD and CTD for sites in the chamber, as observed for these domains in the full-length PhLP2A complex with closed TRiC. How exactly ATP subunit occupancy within and between the rings occurs under physiological conditions of cycling is not completely understood. It will be interesting for future studies to define the precise stoichiometry and allosteric coordination between the TRiC-ATP binding sites, and whether and how these are modulated by the action of PFD, PhLPs as well as binding of folding substrates.

## PhLP2A role in TRiC-mediated substrate folding

PhLP2A contributes to TRiC-mediated substrate folding[25,26], raising the question of how its distinct domains and interactions affect substrate folding. A recent structure of TRiC directly captured from cell lysate showed that it could form a ternary complex with actin and PhLP2A[30]. We added PhLP2A to a reconstituted TRiC-mediated actin folding reaction using purified components and a physiologically relevant system[12]. Briefly, human PFD-bound actin remains in a high protease K sensitive unstructured state (Supplementary Fig. 8a). We used PFD-actin to generate a TRiC-substrate complex followed by the addition of PhLP2A and subsequent incubation with ATP/AlFx (Fig. 6a(i)). CryoEM analyses revealed most TRiC particles adopted the closed state, with ~42% of the particles containing extra-density attributable to PhLP2A or actin encapsulated in each different chamber (Supplementary Fig. 8b). Further classification revealed clear actin and PhLP2A density each occupying two opposite chambers (Fig. 6b). In the closed TRiC chamber, we identified completely folded actin which indicates overall that unstructured actin was transferred from PFD to TRiC and then subsequently folded by TRiC and PhLP2A upon ATP hydrolysis. Of note, the overall architecture of our structure resembles the previously reported TRiC-actin-PhLP2A structure (Fig. 6b)[30], which was a single snapshot structure of an undefined folding state captured from a cell lysate. The consistency between these structures indicates our reconstitution strategy successfully mimics physiologically relevant in vivo TRiC-mediated folding. Since our result is derived from a folding process reconstituted using purified components, it can provide insight into the folding process resulting from a well-defined chemical reaction. Collectively, our structures allow access into the role of PhLP2A in the complete TRiC-mediated actin folding cycle leading to fully folded actin.

We next examined the molecular contacts between PhLP2A in the cis-chamber and actin in the trans-chamber of closed TRiC. Notably, both PhLP2A and actin interact with the positively charged wall hemisphere comprised of residues in CCT 3/6/8 (Fig. 6d). This common binding interface may provide the chemical logic to segregate substrate and cochaperone into two separate chambers. Actin in the chamber appears to be in a fully folded state with its four subdomains (SD1−4) (Fig. 6b(ii, iii), Supplementary Fig. 8e). SD1 and SD2 bind through large surface contact with the CCT1/3/6 wall, while SD3 and SD4 extend towards the center of the chamber without direct TRiC interactions. Our structure corresponds to the previous hydrogen-deuterium exchange and mass spectrometry (HDX-MS) experiment of actin in TRiC, which suggested that SD1 and SD2 is the location protected by TRiC[6]. Notably, the ATP binding pocket is ~20 degrees more open compared to the globular actin structure (PDB 4PKH, Supplementary Fig. 8e)[39], potentially because of the domain-specific constraints from TRiC. On the other side, the overall architecture of PhLP2A is similar to the one in substrate-free TRiC (Fig. 6e, Supplementary Fig. 8f). However, the presence of substrate caused a dramatic change in the NTD H2 to cross the inter-chamber cavity and stretch towards actin (Fig. 6e, Supplementary Movie 4). More importantly, in

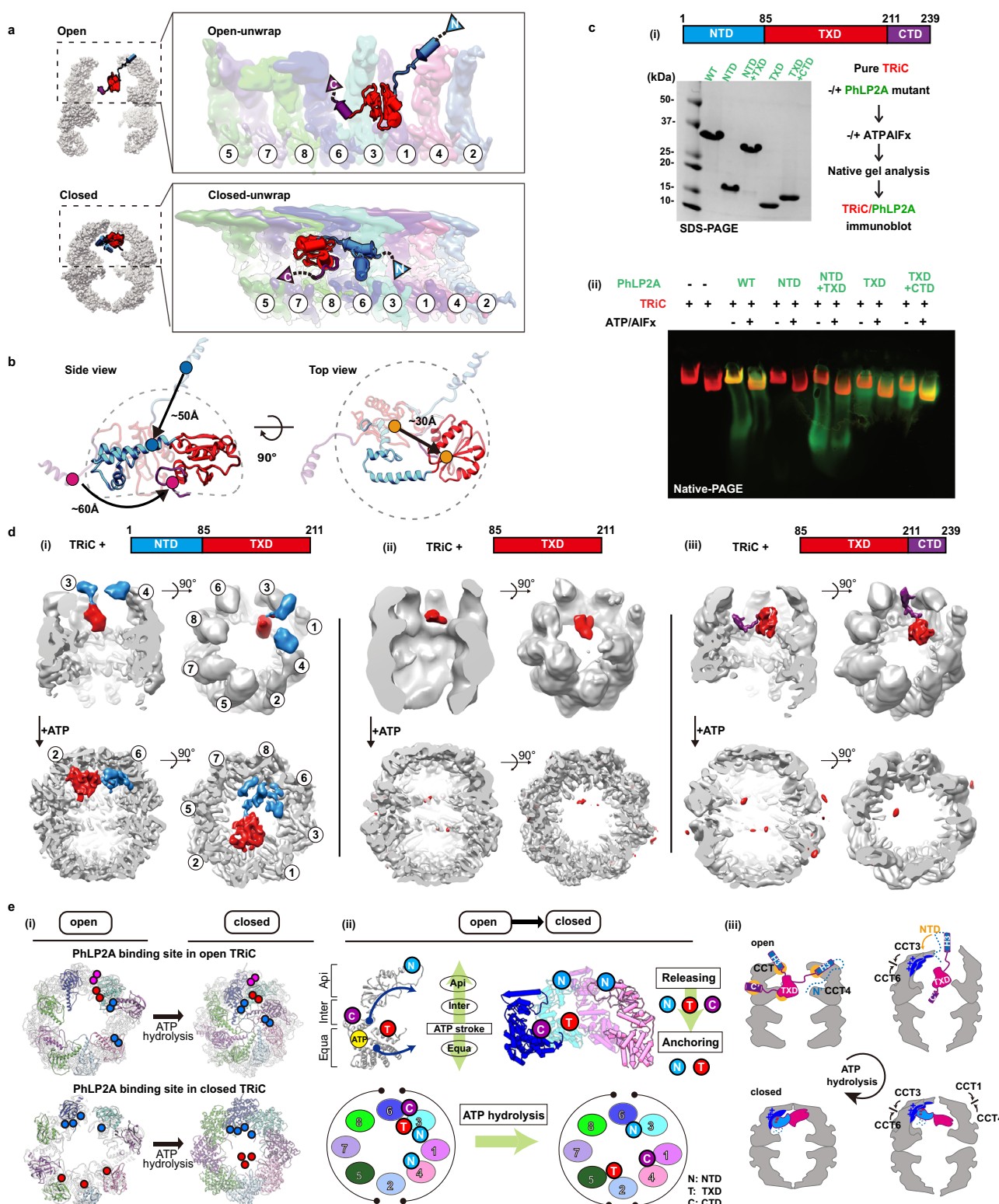

the presence of substrate, the NTD H1 helix was resolved and shown to be in the trans-chamber making direct contact with folded actin (Fig. 6b(ii), c, e, Supplementary Fig. 8f, g, Supplementary Movie 4). This indicates that the presence of bound actin in the trans-chamber of closed TRiC leads to a ~60 Å substrate-induced conformational change in the NTD of PhLP2A (Fig. 6e, Supplementary Movie 4). The cleft between actin SD1 and SD3 contains a hydrophobic groove which is the site that binds the amphipathic helix H1 of PhLP2A (Fig. 6f, Supplementary Fig. 8h, Supplementary Movie 4). Of note, this hydrophobic groove is an interaction hotspot for regulators of actin polymerization[40] as well as a mutation hotspot affecting actin folding (Fig. 6g, Supplementary Fig. 8i)[41]. Taken together, our structural analysis suggests that TRiC and PhLP2A cooperatively orchestrate actin folding. The TRiC chamber wall provides a rigid molecular template to bring the actin subdomains together to establish the topology and H1 of PhLP2A, as an actin specific contact, of PhLP2A serves to seal the exposed hydrophobic surface between SD1 and 3 which may facilitate the formation of this actin lobe and assist folding.

**Fig. 5 | Domain-wise characteristics of PhLP2A in relationship with TRiC.**
**a** Global rearrangement of PhLP2A in the TRiC folding chamber in the unwrapped view. (Top) Unwrapped view of PhLP2A in which NTD is in the expanded form in TRiC in the open conformation. (Bottom) Unwrapped view of PhLP2A in the closed conformation of TRiC. Note that PhLP2A NTD is compacted and constrained.
**b** Conformational and orientation changes of PhLP2A in the open or closed TRiC folding chamber. PhLP2A NTD undergoes an orientation change of about ~50 Å from extended outward to bent inward, closer to the TXD. The TXD is lifted about ~30 Å from the equator to the apical contact through flipping but retains its conformation. The CTD moves ~60 Å following the movement of TXD. **c** (i) (Top) A schematic diagram of PhLP2A domain. (Left) SDS-PAGE of PhLP2A wild-type (WT) and four truncated mutant constructs of PhLP2A: NTD (aa 1–84), CTD truncation (aa 1–211, NTD + TXD), TXD (aa 85–211), NTD truncation (aa 85–239, TXD + CTD). (Right) Schematic of the binding assay. (ii) Binding assay between TRiC and PhLP2A truncation mutants with or without ATP/AlFx. This experiment is performed in

duplicate. **d** CryoEM structures of three truncated mutants inside the TRiC in the open state and after ATP hydrolysis. (i) TRiC with NTD + TXD: attributable NTD density at the apical domain with TXD and NTD + TXD orients like WT PhLP2A in closed TRiC. (ii) TRiC with TXD: attributable density of TXD at low resolution at the equator, but no attributable density in closed TRiC. (iii) TRiC with TXD + CTD: CTD anchor shows high-resolution features like WT PhLP2A between CCT3/6 and TXD close to CCT3, but no attributable density in closed TRiC. **e** (i) Schematic diagram of PhLP2A domain-wise interaction with the TRiC chamber in open (left) and closed conformation (right). Colored circles indicate interacting residues in open TRiC and closed TRiC and their movements during the ATP cycle. Each PhLP2A domain is color-coded as in **c**. (ii) ATP hydrolysis event in CCT subunits cascades the releasing and re-anchoring process of PhLP2A. N: NTD, T: TXD, C: CTD. (iii) Diagram of releasing of PhLP2A upon ATP-dependent cycle of TRiC. Source data are provided as a Source Data file.

## Evolutionary conservation in the PhLP superfamily

Our analysis indicates each domain of PhLP2A plays a singular yet synergistic role in this cochaperone association with and modulation of TRiC. Since PhLPs have diverged into distinct homologs, we next considered the implications of our conclusions to understand the activity of this superfamily and the sources of functional difference[18,19]. A phylogenetic tree constructed using the full-length sequences of 71 eukaryotic PhLPs revealed the evolutionary hierarchy among PhLP family members (Fig. 7a). The simple PhLP system from unicellular organisms (e.g., PLP1 and PLP2 in yeast), diverged into diverse PhLP1-3 subfamilies in multicellular organisms in addition to the phosducin (PDC) family. Human PhLP3 is closest to yeast PLP1 and human PhLP2 is closest to yeast PLP2. We also find the PhLP1-3 branches preceded the appearance of phosducin (PDC), which was historically identified first. PhLP1, 2A and 3 are ubiquitously expressed in a wide variety of tissues[21,42] while PDC and PhLP2B are each expressed exclusively in the retina and the reproductive tissues[18,43,44]. This suggests PDC and PhLP2B diverged later to function in a tissue-specific manner.

While all PhLPs share a similar domain architecture, we observe varying levels of conservation in different domains (Fig. 7b). The domain elements that contact both open and closed TRiC, namely H3 of the NTD and TXD, are highly conserved across PhLPs particularly the TRiC interacting positive patch on H3 and the hydrophobic surface of TXD (Fig. 7c, Supplementary Fig. 9a). This pattern of conservation suggests that all PhLP family members interact with open and closed TRiC through similar domain-wise contacts as described here for PhLP2A. Indeed, previous reports indicate that PhLP1-3 can all bind TRiC[26,43,45,46]. In contrast, the TRiC interacting residues are significantly diverged in PDC (Supplementary Fig. 9a), despite having ~28% sequence identity with PhLP2A; this explains previous findings that PDC does not bind TRiC[45,47].

Interestingly, both the NTD and the CTD highly varied among PhLP family members (Fig. 7b, Supplementary Fig. 9a–c). For instance, the sequence of PhLP2A contacting TRiC-bound actin in our analysis maps to a variable NTD region (Supplementary Fig. 9b). Such sequence variation may confer PhLPs with the ability to contact different folding substrates within the TRiC chamber. For instance, H1 in the NTD of PhLP2B also has a -WNDIL- actin-binding motif, similar to that identified in PhLP2A in this study and previously[30]. This sequence is weak in other PhLP variants. On the other hand, the structure of a PDC-Gβ complex identified a -GVI- Gβ binding motif in PDC that is also highly conserved in PhLP1. In contrast, this motif is absent in PhLP2A and PhLP2B, which cannot interact with Gβ[20]. It thus appears that PhLPs isoforms share conserved TRiC binding elements but evolved distinct substrate-binding elements which determine PhLPs action within the TRiC folding chamber.

## Discussion

TRiC functions in a cooperative network with PhLPs[16]. Our analysis shows that PhLP2A can bind to both open and closed TRiC (Figs. 1, 4).

In both conformations, PhLP2A binds inside the chamber but undergoes significant changes upon TRiC closure and in the presence of substrate (Figs. 5 and 7d(i)). In the open state, the PhLP2A NTD adopts an H3 extended conformation to apical domains either CCT3 or 4. ATP-dependent TRiC closure encapsulates substrate and induces a large reorientation of PhLP2A domains, with reorienting the NTD H3 and shifting its binding sites inside the closed TRiC chamber. Within the chamber, PhLP2A becomes compacted with the highly negatively charged H2 and part of H3 associating with the positive hemisphere within the TRiC chamber. The presence of substrate in the opposite closed TRiC chamber induces a further conformational change of PhLP2A, specifically in H1 and H2 of the NTD, whereby H2 traverses the chamber positioning H1 to directly bind the substrate actin. H1 of PhLP2A masks a hydrophobic groove around two helices (aa 137–145 and 340–350) of encapsulated actin (Fig. 6f, Supplementary Fig. 8h). Our structure suggest PhLP2A H1 stabilizes an exposed hydrophobic core of actin intermediates that are folding within the TRiC chamber. While future experiments should delve into the mechanism of how PhLP2A assists folding, it is noteworthy that mutation in actin residues 340–344, *i.e.* corresponding to the PhLP2A H1 binding site, significantly affects actin folding[41,48]. Furthermore, previous studies observed that the folding of SD1 is rate-limiting for TRiC-mediated actin folding[6]. Taken together, our study provides mechanistic insight into the molecular interplay between TRiC and PhLP2A to orchestrate a hydrophobic collapse that drives the folding of the substrate.

Our finding that PhLP2A H3 shares a TRiC binding site with PFD and their binding is mutually exclusive (Fig. 3, Supplementary Fig. 4a, f) suggests PhLP2A coordinates association between PFD and TRiC. How these observations relate to the function of PhLPs in the TRiC-mediated folding reaction remains to be investigated. Our biochemical and structural experiments support a compelling hypothesis whereby the overlapping and non-overlapping chaperonin binding interfaces in PFD and PhLPs establish a directional TRiC cochaperone network. We propose that, following PFD delivery of its bound substrate to TRiC, it can be displaced from TRiC by PhLP2A binding. Following ATP hydrolysis, bound PhLP2A remains within the TRiC chamber to facilitate substrate folding (Fig. 7d(ii)). This proposed model should be critically tested in future experiments, but could explain old observations whereby addition of high levels of PhLP3 to an in vitro translation of actin in rabbit reticulocyte lysate led to the accumulation of actin-bound PFD[26] and inhibited TRiC-mediated folding of newly synthesized actin[45]. Based on our model, we propose the high amounts of PhLPs added to the system saturate the PFD binding sites in TRiC, precluding substrate transfer to the chaperonin.

Recently, Han et al.[31], conducted an investigation on TRiC-Plp2 from yeast, which exhibits homology to the human TRiC complex with PhLP. While we did not explore the ternary complex of actin in open TRiC-PhLP2A conformation, Han et al., resolved a ternary complex structure of TRiC-Plp2-actin in nucleotide free and ATP bound open

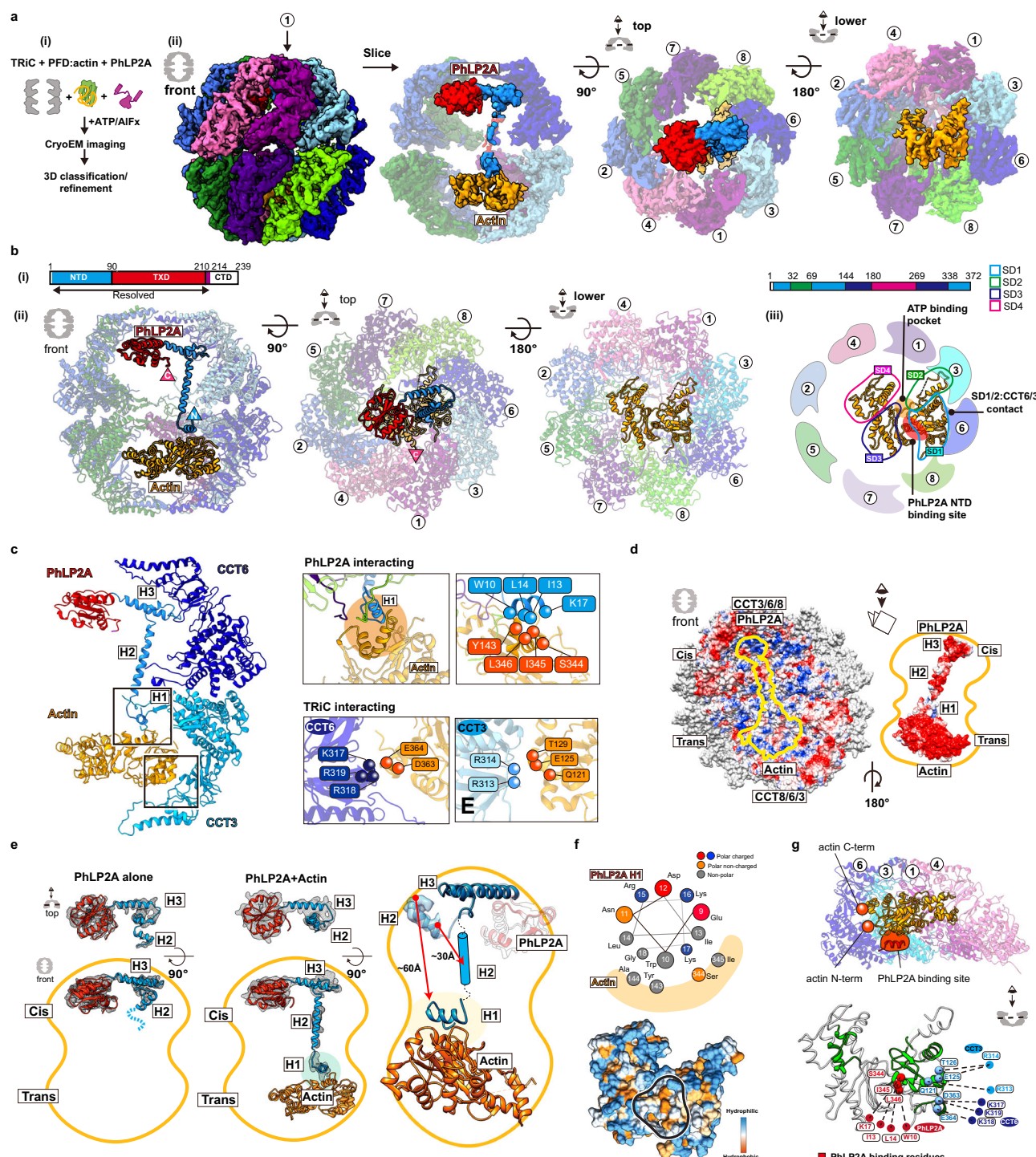

**Fig. 6 | CryoEM structure of closed TRiC with folded actin and PhLP2A.**
**a** CryoEM structure of closed TRiC with folded actin and PhLP2A in each chamber.
(i) Sample preparation scheme for the substrate-cochaperone TRiC. (ii) CryoEM
map of folded actin and PhLP2A encapsulated in closed TRiC. The density of H2 of
PhLP2A is low-pass filtered and depicted at σ = 1.4 from the density map of the full
complex. CCT1 is indicated by an arrow. **b** Atomic model of closed PhLP2A-actin-
TRiC (i) Schematic of the model of PhLP2A. (ii) Slice views of PhLP2A and folded
actin encapsulated inside the closed TRiC. (iii) Summary of actin features inside the
folding chamber. SD1 is the major binding site with the intermediate domain of
CCT3/6. The PhLP2A binding site is between SD1 and SD3 (indicated by a red circle),
and the ATP binding pocket is on the opposite side, between SD2 and SD4 (indi-
cated by an orange marker). **c** Detailed interactions between PhLP2A, CCT subunits,
and folded actin. (Left) Interactions between the PhLP2A NTD or CCT3 and actin.

(Right) (Top) H1 of PhLP2A and CCT3 show direct interaction with actin, forming a
local hydrophobic interaction network. Interacting residues are represented as
balls. (Bottom) Interacting residues between actin and CCT3/6 are shown as balls.
**d** Electrostatic surface of the closed TRiC chambers (left), PhLP2A and actin (right).
**e** The comparison between PhLP2A with and without actin in the closed folding
chamber. Conformational changes of PhLP2A induced by the encapsulated sub-
strate are represented. **f** The helix wheel plot and hydrophobic surface of encap-
sulated actin showing the actin-PhLP2A contact sites. **g** (Top) The slice view of the
actin-TRiC contact site. CCT4/1/3/6 and actin are shown. N- and C-terminus of actin
are represented as balls and PhLP2A binding site is colored in red. (Bottom) The
folding defect mutant residues colored in green on the encapsulated actin
structure[34]. Red balls indicate the residues interacting with PhLP2A while blue balls
represent CCT interacting residues.

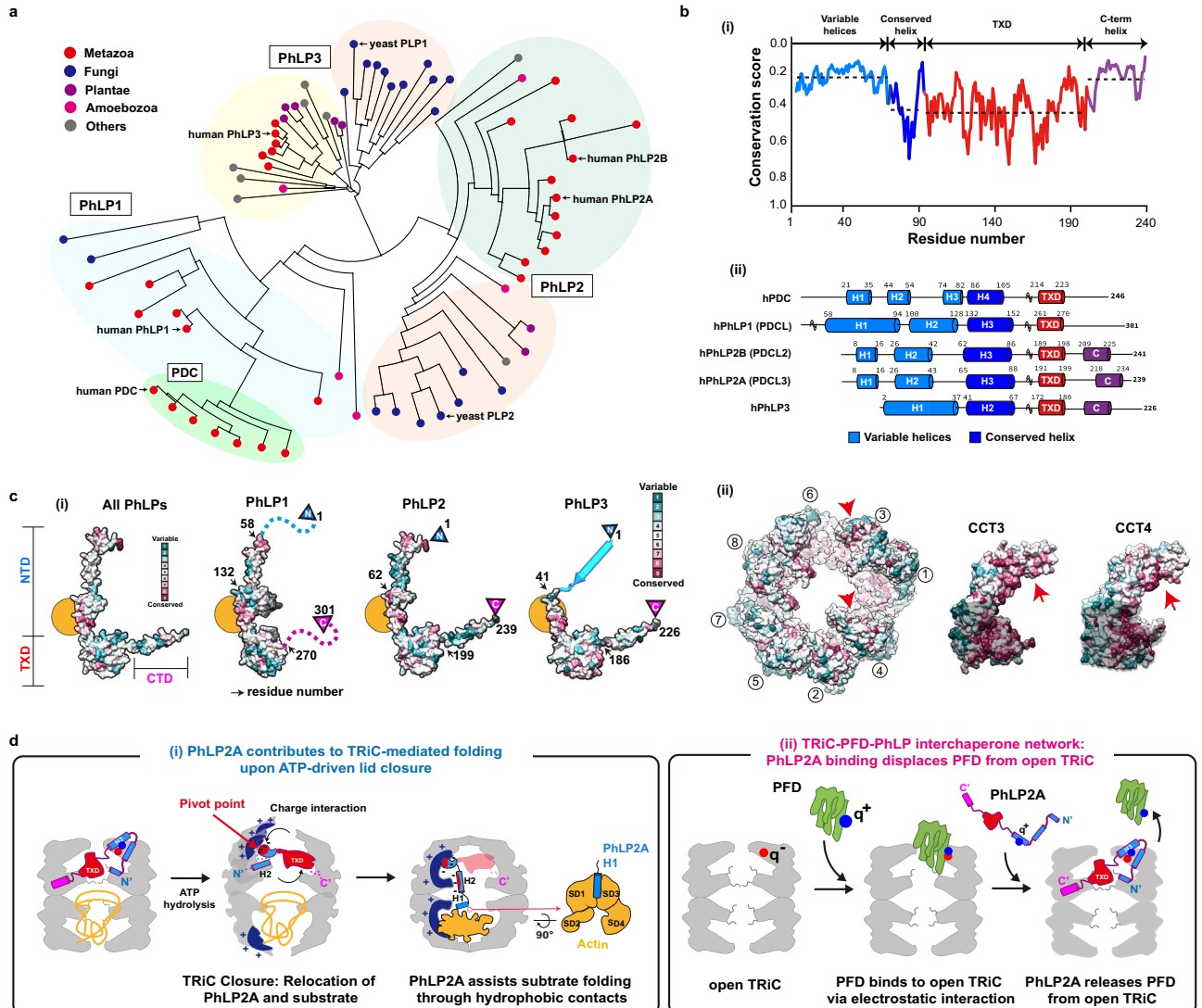

**Fig. 7 | Conservation on phosducin-like protein family and the proposed mechanism of chaperone-in-chaperonin mediated folding cycle. a** Phylogenetic tree of the PhLP family. The kingdom of each branch and each human subtype (PhLP1, PhLP2A, PhLP2B, PhLP3, PDC) are indicated in the phylogenetic tree. **b** (i) The domain-wise conservation score of PhLPs. Each domain is divided according to human PhLP2A domain features. The dotted line indicates the average value of the conservation score for each domain. (ii) The secondary structures of 5 human PhLPs based on AlphaFold predictions. **c** Conservation scores of human PhLPs and TRiC. (i) Residue conservation of human PhLPs. The yellow circle indicates the conserved surface in H3 helix. (ii) Residue conservation among CCT subunits, and CCT3, CCT4 is shown. **d** Proposed mechanism of PhLP2A-mediated substrate folding. (i) PhLP2A contributes to TRiC-mediated substrate folding. When ATP hydrolysis occurs, the positively charged surface inside the TRiC chamber is formed through CCT3/6. The negatively charged NTD of PhLP2A acts as a pivot point to induce 90 degrees relocation of PhLP2A in the cis-ring. The substrate migrates to the trans-ring. Once the trans chamber is occupied by the substrate, H1 and H2 of PhLP2A traverse the two chambers. H2 penetrates the middle of the two rings, and H1 makes direct contact with the substrate. A hydrophobic network between the PhLP2A H1 helix and the substrate might assist the substrate folding. (ii) PhLP2A binding displaces PFD from open TRiC. When PhLP2A binds to TRiC, its NTD displaces PFD by competing with the same binding sites. PhLP2A also inhibits further PFD interactions and possibly prevents substrate overloading.

conformations. In the snapshot structure of closed TRIC, Han et al. observe folded actin in one chamber, and Plp2 in the opposite chamber with an overall architecture and specific CCT interaction that are almost identical to what we identify in the human system in this study. However, unlike our study, they do not find the extended helices of Plp2 making direct contact with actin. It is noteworthy that the open conformation of the TRiC-Plp2 complex in that study shows one Plp2 molecule in the cis-TRiC chamber and additional densities present in the trans-TRiC chamber, which was suggested as a substrate bound to specific CCT subunits. Han et al. also discussed significant density in the septum as a tangle of CCT tails, although we propose this septum density may also contain substrate tethered by CCT tails, as supported by their own crosslinking mass analysis. Of special note Han et al. utilized chemically denatured actin throughout their study, while our

approach involves using copurified actin-PFD to introduce unstructured actin into the TRiC system. In a separate study by Gestaut et al.[17], it was shown that substrates bind to PFD in an unstructured state, which can undergo folding upon delivery to TRiC in the presence of ATP. This discrepancy in experimental design may influence the determination of the substrate binding location within the open TRiC chamber. On the other hand, it has been observed that PFD delivers another substrate, tubulin, deep inside the TRiC chamber, compatible with the density in septum that was attributed as substrate[12]. Consequently, the mechanism of PFD-mediated actin delivery to open TRiC remains uncertain and necessitates further investigation. Addressing this aspect in future studies is essential to shed light on the intricacies of chaperone-assisted protein folding and the specific recognition of substrates.

TRiC facilitates the folding of ~10% of the proteome and diversifies the strategies to support folding for many different types of protein substrates. There is increasing evidence that the unique interior of the TRiC chamber plays a multifaceted and complex role directly in the folding of the bound substrate[17,30,31]. The heterooligomeric nature of TRiC is considered as a key to diversifying the recognition of distinct motifs in its diverse substrates[6,7,49], as well as promoting their folding within the closed chamber[6,12]. Our analysis of PhLP family evolution suggests the modality of binding of family members to the TRiC chamber is largely conserved, whereas the substrate binding elements within the NTD, and perhaps the CTD, have diversified to assist folding to different substrate proteins. Indeed, different PhLP family members have different substrate specificity and function[23,24] through substrate-specific sequences in highly diverged NTD[25,26]. PhLPs are only found in eukaryotes and their advent may be linked to the increase in proteomic complexity[50]. We propose that the combinatorial usage of PhLPs as substrate-specific cochaperones expands the ability of TRiC to fold different substrates. Notably, the activity of PhLP can be regulated by phosphorylation[51,52], adding another layer of regulation to TRiC action in the cell. Our work uncovers another layer of complexity in the regulation of TRiC cochaperone networks mediating cellular folding.

## Methods

### Cloning

PhLP2A and fragments were cloned into the pST39 backbone using traditional cloning methods. PhLP2A was amplified from a human cDNA library using 5' primer Cagtt Tctagatttgtttaacttaagaaggagatatacatatgcaagatcccaatgaagatacag and 3' primer aattaggatccttatcagtgatggtggtgatggtgaaagctatttattgaatatttctctc. PhLP2A fragments were cloned by amplifying from above with 5' PhLP2A primer and 3' AAT TAG GAT CCT TAT CAG TGA TGG TGG TGA TGG TGA GTT GCT TTC CAC TCA GCC for N-term fragment, 5' CAG TTT CTA GAT TTG TTT AAC TTT AAG AAG GAG ATA TAC ATA TGA AAC TGA AGA ATA AAT TCG G and 3' PhLP2A for TXT-C term, 5' PhLP2A and 3' AAT TAG GAT CCT TAT CAG TGA TGG TGG TGA TGG TGG TTT TCC TCC AGG TCT GTC for N + TXT fragment, and 5' CAG TTT CTA GAT TTG TTT AAC TTT AAG AAG GAG ATA TAC ATA TGA AAC TGA AGA ATA AAT TCG G and 3' AAT TAG GAT CCT TAT CAG TGA TGG TGG TGA TGG TGG TTT TCC TCC AGG TCT GTC for TXT alone. All variants were cloned into pST39 using XbaI and BamHI.

### Protein expression and purification

**TRiC alone.** High Five insect cells (1 L) were co-infected with rBVs encoding his-tagged TRiC with CCT1 tagged with GFP. Cells were incubated at 27 °C for ~72 h, pelleted by centrifugation at $500 \times g$ for 10 min, and resuspended in TRiC lysis buffer (100 mM HEPES pH 7.4, 50 mM NaCl, 20 mM imidazole, 10% glycerol, 5 mM PMSF) supplemented with benzonase (Sigma-Aldrich-Aldrich, E1014) (1,000 units) and a protease inhibitor cocktail [Roche]). Cells were lysed using Dounce homogenization and debris were cleared by ultracentrifugation at $50,000 \times g$ at 4 °C for 40 min. Cleared supernatant was passed over nickel resin and washed with column wash buffer (50 mM HEPES pH 7.4, 50 mM NaCl, 5 mM MgCl₂ 20 mM imidazole, 10% glycerol) with an additional 250 mM NaCl, column wash buffer +1 mM ATP, column wash buffer + an additional 500 mM NaCl, and finally column wash buffer alone. Nickel-bound protein was eluted with an elution buffer (50 mM HEPES pH 7.4, 50 mM NaCl, 5 mM MgCl₂ 400 mM imidazole, 10% glycerol). Protein-containing fractions were pooled and passed over a heparin column equilibrated with MQA buffer (50 mM HEPES pH 7.4, 50 mM NaCl, 5 mM MgCl₂, 0.5 mM EDTA, 1 mM DTT, 10% glycerol). Protein was eluted with a linear gradient of 20% to 100% MQB buffer (50 mM HEPES pH 7.4, 1 M NaCl, 5 mM MgCl₂, 0.5 mM EDTA, 1 mM DTT, 10% glycerol). TRiC-containing fractions were pooled and diluted with MQ buffer (50 mM HEPES pH 7.4, 5 mM MgCl₂, 1 mM DTT, 10% glycerol) to remove excess NaCl. Pooled TRiC was loaded onto a MonoQ ion exchange column and eluted with a 200 ml linear gradient of 0% to 100% MQB. TRiC-containing fractions were pooled, concentrated with a 100 kDa MWCO Centricon device, and passed over a Superose-6 size exclusion column equilibrated with MQA. TRiC-containing fractions were pooled, concentrated, and snap-frozen in liquid nitrogen for long-term storage.

**Human Prefoldin.** Human Prefoldin was expressed and purified as described previously[17]. All subunits were co-expressed using baculovirus in High Five insect cells. The Prefoldin complex was isolated by affinity (Nickel-NTA), anion exchange (MonoQ 10/100), and gel filtration (Superdex 200) chromatography. At each step, fractions containing Prefoldin were identified by SDS-PAGE. Final protein was concentrated to ~100 μM using an Amicon Ultra 30 kDa MWCO, and 50% glycerol was added to yield a final concentration of 10%. Protein was aliquoted, snap-frozen, and stored at −80 °C.

**Human PhLP2A.** Plasmids were transformed into BL21 Rosetta2 pLysS and induced to express O/N at 17 °C. Cells were resuspended in lysis buffer 50 mM HEPES pH 7.4, 600 mM NaCl, 20 mM imidazole, 5 mM PMSF) supplemented with benzonase (Sigma-Aldrich-Aldrich, E1014) (1000 units) and a protease inhibitor cocktail [Roche]) and lysed using an emulsiflex. The lysate was cleared by centrifugation at 20,000*g for 30 min. Cleared lysate was purified by passing over nickel resin and washing with lysis buffer followed by column buffer (lysis buffer lacking protease inhibitors). Protein was eluted using column buffer +400 mM imidazole. Fractions containing protein were concentrated using Amicon ultra 3 kD MWCO concentrators and ran over an SDX200 equilibrated with 50 mM HEPES pH 7.4, 100 mM NaCl, 1 mM DTT. Fractions containing PhLP2a protein were identified by SDS-PAGE and protein was concentrated using an Amicon Ultra 3 kD MWCO. 50% glycerol was added to a final concentration of 5% to protein, and the protein was aliquoted, snap frozen, and stored at −80 °C.

**Human PFD-β-actin.** All subunits of *Hs*PFD and β-actin were co-expressed using baculovirus in High Five insect cells (twice as much virus was used for β-actin compared to *Hs*PFD subunits). Cells were resuspended in lysis buffer (100 mM HEPES pH 7.4, 100 mM NaCl, 15 mM imidazole, 5 mM PMSF) supplemented with benzonase (Sigma-Aldrich-Aldrich, E1014) (1,000 units) and a protease inhibitor cocktail [Roche]) and lysed using Dounce homogenization. Lysate was cleared by ultracentrifugation at 40,000 * g for 30 min. Cleared lysate was passed over nickel resin, washed with column buffer (100 mM NaCl, 50 mM HEPES pH 7.4) and eluted with column buffer +400 mM imidazole. Fractions containing protein were passed over a MonoQ 10/100 anion exchange column equilibrated with buffer A (50 mM HEPES pH 7.4, 50 mM NaCl) washed with buffer A until UV baselined, and then eluted with a 160 ml gradient to 60% buffer B (50 mM HEPES pH 7.4, 1 M NaCl). Fractions containing PFD- β-actin were identified by SDS-PAGE, concentrated to ~1 ml, and ran over an SDX200 column. Fractions containing PFD-βActin were again identified by SDS-PAGE, concentrated, 50% glycerol added to 10%, aliquoted, snap-frozen and stored at −80 °C

### Native and SDS PAGE analysis

**SDS gel analysis for Prefoldin-PhLP2A fragments.** PhLP2A fragments were diluted to 40 μM and 15 μL of the sample was mixed with 5 μL of 4XPSB (200 mM Tris pH 6.8, 8% SDS, 40% glycerol, 20% b-mercaptoethanol, bromophenol blue to color) was added to 24 μL of the sample, and 10 μl of each sample was separated in 15% SDS-PAGE gels. The sample was stained by Coomassie for detection

**PhLP2A fragment binding to TRiC Native PAGE analysis.** Proteins were mixed to reach a final concentration of 0.5 μM bovine TRiC and 5 μM PhLP2A fragment after the addition of either H₂O or ATP/AlFx

(1 mM ATP, 1 mM AlNO$_3$, 6 mM NaF). Samples were separated on a 5% acrylamide gel containing 80 mM HEPES pH 8 and 1 mM MgCl$_2$. The running buffer consisted of 80 mM HEPES pH 8, 1 mM MgCl$_2$, 1 mM DTT, and 1 mg/ml L-cysteine. Proteins were transferred to nitrocellulose and immunoblotted with anti-His (MA1-21315 Thermo Fisher Scientific) 1:2000, and anti-CCT4 Rabbit Invitrogen 1:2500.

**Proteinase K assays**
PFD-actin and G-actin were brought to a final concentration of 0.5 µM in 1XATPase buffer (50 mM HEPES, 50 mM KCl, 5 mM MgCl$_2$, 10% glycerol, 1 mM DTT). Samples were brought to 37 °C and 1 mM ATP was added to both samples. After 30 min, samples were brought to room temperature. 15 µL was removed from each sample for zero-minute time points, and digestion was initiated by the addition of proteinase K to a final concentration of 2 ng/µL. Timepoints were taken at 2, 4, 8 and 16 min by adding 15 µL sample to 2 µL of PMSF, followed by 6 µL 4XPSB. Samples were analyzed by SDS-PAGE followed by immunoblot with anti-Actin (JLA20 – https://dshb.biology.uiowa.edu/JLA20).

**Native PAGE mobility shift assay**
AF-568 labeled PhLP2A and Human TRiC were combined in ATPase buffer (30 mM Tris pH 7.4, 120 mM KCl, 5 mM MgCl2, and 1 mM dithiothreitol freshly added). This mixture was subjected to serial dilution, and then mixed in a 1:1 ratio with TRiC, resulting in a final concentration of 0.25 µM TRiC and PhLP2A ranging from 0.01 to 10 µM. The binding process was allowed to reach equilibrium for approximately 30 min at room temperature. Subsequently, the samples were subjected to clear Native PAGE (either 4–12% or 4–16%). In-gel fluorescence analysis was performed at a wavelength of 610 nm to assess TRiC-bound PhLP2A. The relative Kd values were determined through three independent repetitions of the analysis. To assess the ability of either PhLP2A binding to displace PFD binding from TRiC, or PFD binding to displace PhLP2A binding from TRiC, competition assays were performed in 1XATPase buffer. In these assays TRiC at 0.25 nM was preincubated with either PFD labeled with Alexa647 or PhLP2A labeled with Alexa568 at 1 uM. Serially diluted unlabeled PhLP2A (0.01–10 µM) were applied to the mixture of GFP-TRiC (0.25 µM) and AF-647 labeled PFD (1 µM) and then protein complex were separated in Native PAGE 4–16% (Invitrogen BN1004BOX). In-gel fluorescence analysis was performed at a wavelength of 670 nm and 526 nm to assess the TRiC-bound PFD and displaced PFD. In the same manner, serially diluted unlabeled PFD (0.01–10 µM) were applied to the mixture of GFP-TRiC (0.25 µM) and AF-568 labeled PhLP2A (1 µM) and then protein complexes were separated in Native PAGE 4–16% (Invitrogen BN1004BOX). In-gel fluorescence analysis was performed at a wavelength of 610 nm and 526 nm to assess the amount of TRiC-bound PhLP2A and free PhLP2A.

**Crosslinking-Mass Spectrometry**
**Crosslinked sample preparation.** *Hs*TRiC and PhLP2A were mixed to a final concentration of 1 µM and 4 µM respectively in 1X ATPase buffer (30 mM HEPES pH 7.4, 100 mM KCl, 5 mM MgCl$_2$, 1 mM DTT, 10% glycerol). Freshly dissolved heavy/light (H12/D12) disuccinimidyl suberate (DSS) (Creative Molecules, 001 S) in DMSO (50 mM) was added to a final concentration of 1 mM to the protein mixture and incubated at room temperature for 1 h. The crosslinking reaction was quenched by the addition of Tris pH 7.4 to ~100 mM and incubation for 30 min at RT. The sample was snap-frozen for further processing as described previously[12]. Briefly, processing steps included reduction of disulfide bonds using tris(2-carboxyethyl) phosphine, alkylation of free thiol groups on cysteines with iodoacetamide, and proteolysis with endoproteinase Lys-C and trypsin. The resulting peptides were fractionated using peptide-level size exclusion chromatography (SEC). The detailed results are shown in Supplementary Data 1.

**Liquid chromatography-tandem mass spectrometry (LC-MS/MS).** The analysis was performed on an Easy nLC-1200 HPLC system coupled to an Orbitrap Fusion Lumos mass spectrometer (both Thermo-Fisher Scientific). Single injections of three SEC fractions were performed. Peptides were separated by reversed-phase chromatography on an Acclaim PepMap RLSC C18 column (250 mm × 75 µm, ThermoFisher Scientific) at a flow rate of 300 nl/min. The mobile phase gradient was 11 to 40% B in 60 min, with mobile phases A=water/acetonitrile/formic acid (98:2:0.15, v/v/v) and B=acetonitrile/water/formic acid (80:20:0.15, v/v/v). MS/MS data were acquired in the data-dependent acquisition top speed mode with a cycle time of 3 s. Precursor and fragment ion spectra were acquired in the Orbitrap at 120,000 and 30,000 resolutions, respectively. Precursor ions with a charge state of +3 to +7 were fragmented in the linear ion trap at a normalized collision energy of 35%. Dynamic exclusion was enabled for 30 s after one sequencing event.

**Data analysis.** MS/MS spectra were analyzed using xQuest, version 2.1.5 (available from https://gitlab.ethz.ch/leitner_lab/xquest_xprophet). The database contained the entries of all human TRiC subunits and PhLP2A. No other contaminant proteins were observed at relevant levels. Search parameters for xQuest included: enzyme = trypsin, maximum number of missed cleavages = 2, carbamidomethylation of Cys as fixed modification, oxidation of Met as variable modification, Lys and protein N terminus as crosslinking sites, mass error tolerances of ±15 ppm at the MS1 level and ±20 ppm at the MS2 level. Search results were further filtered with stricter mass tolerances depending on the dataset, and identifications were required to have an xQuest delta score <0.9, a TIC score >0.1, and a minimum of four bond cleavages overall or three consecutive bond cleavages per peptide. A parallel search against the reversed and shuffled sequences of the database entries revealed no decoy hits fulfilling the search and filter criteria, suggesting the false discovery rate is close to 0% for the selected score thresholds. Mapping of XLs was performed using xiVIEW (Graham, M., Combe, C. W., Kolbowski, L. & Rappsilber, J. xiView: A common platform for the downstream analysis of Crosslinking Mass Spectrometry data. BioRxiv, doi: 10.1101/561829).

**CryoEM specimen preparation and data collection**
**PhLP2A-TRiC.** 2 mg/ml of purified TRiC was incubated at RT with purified PhLP2A in 1:4 molar ratio for 2 h. Then, 3 µL of PhLP2A-TRiC sample was applied to 200-mesh R1.2/1.3 holey-carbon grids (Quantifoil) coated with Poly-L-lysine and blotted for 3.5 s and vitrified using Vitrobot Mark IV (Thermo Fisher Scientific, CMCI in Seoul). 15,075 movies were collected on a Titan Krios (Thermo Fisher Scientific) equipped with a K3 BioQuantum detector with 20 eV energy filter slit (Gatan) in CDS mode. The detailed imaging conditions are shown in Supplementary Table 1.

**PhLP2A-TRiC with ATP/AlFx.** 0.8 mg/ml of purified TRiC was mixed with PhLP2A in 1:4 molar ratio for 30 min. Then, 1 mM ATP, 1 mM Al$_3$(NO$_3$)$_3$, 6 mM NaF, 10 mM MgCl$_2$ 50 mM KCl were added and incubated for 1 h at RT. 3 µL of PhLP2A-TRiC ATP/AlFx sample was applied to glow discharged cryoEM grids (Quantifoil R1.2/1.3 200 Cu). Samples were vitrified using Vitrobot Mark IV. 1,368 movies were collected on a Glacios (Thermo Fisher Scientific) equipped with the Falcon 4 (Thermo Fisher Scientific) detector. The detailed imaging conditions are shown in Supplementary Table 1.

**PFD-PhLP2A-TRiC.** 2 mg/ml of purified TRiC was sequentially incubated with purified PFD and PhLP2A in 1:2: ~1 for each 20 min at RT with 1 mM AMP-PNP. Then, an aliquot of 3 µL of this sample was applied to 200-mesh R1.2/1.3 holey-carbon grids (Quantifoil) blotted for 3.5 s and vitrified using Vitrobot Mark IV (Thermo Fisher Scientific, CMCI in Seoul). 1,270 movies were collected on a Glacios (Thermo

Fisher Scientific) equipped with the Falcon 4 (Thermo Fisher Scientific) detector. The detailed imaging conditions are shown in Supplementary Table 1.

**Truncated PhLP2A-TRiC.** 2 mg/ml of purified TRiC was incubated with each purified truncated PhLP2A mutant (TXD, NTD-TXD, TXD-CTD) in 1:2 molar ratio for 20 min at RT. Then, an aliquot of 3 µL of this sample was applied to 200-mesh R1.2/1.3 holey-carbon grids (Quantifoil) blotted for 3.5 s and vitrified using Vitrobot Mark IV (Thermo Fisher Scientific, CMCI in Seoul). 2,199 movies for NTD-TXD, 352 movies for TXD and 339 movies for TXD-CTD were collected on a Glacios (Thermo Fisher Scientific) equipped with the Falcon 4 (Thermo Fisher Scientific) detector. The detailed imaging conditions are shown in Supplementary Table 1.

**Truncated PhLP2A-TRiC with ATP/AlFx.** To prepare the sample of TRiC in the presence of 1 mM ATP/AlFx, 0.8 mg/ml of purified TRiC was incubated with each purified truncated PhLP2A mutant (TXD, NTD-TXD, TXD-CTD) in 1:2 molar ratio for 20 min, then 1 mM ATP, 1 mM $Al_3(NO_3)_3$, 6 mM NaF, 10 mM $MgCl_2$ and 50 mM KCl were added and incubated for 1 h at RT. Then, 3 µL of truncated PhLP2A-TRiC sample was applied to 200-mesh R1.2/1.3 holey-carbon grids (Quantifoil) blotted for 3.5 sec and vitrified using Vitrobot Mark IV (Thermo Fisher Scientific, CMCI in Seoul). 2,014 movies for NTD-TXD, 516 movies for TXD, and 644 movies for TXD-CTD were collected on a Glacios (Thermo Fisher Scientific) equipped with the Falcon 4 (Thermo Fisher Scientific) detector. The detailed imaging conditions are shown in Supplementary Table 1.

**PhLP2A-β-actin-TRiC with ATP/AlFx.** 0.8 mg/ml of purified TRiC was mixed with co-purified PFD- β-actin and PhLP2A in a 1:2:2 molar ratio. They were incubated for 20 min and 1 mM ATP, 1 mM $Al_3(NO_3)_3$, 6 mM NaF, 10 mM $MgCl_2$ and 50 mM KCl were added followed by incubation for 30 min at 37 °C. 3 µL droplets of samples were then applied to Vitrobot Mark IV (Thermo Fisher Scientific, CMCI in Seoul) and blotted for 3.5 s. After plunge freezing, 3635 micrographs were collected on Glacios (Thermo Fisher Scientific) equipped with the Falcon 4 (Thermo Fisher Scientific) detector.

### Data processing and 3D refinement

All image processing was done in RELION 4.0[53] and cryoSPARC v3.3[54]. Computing resources were utilized at CMCI at Seoul National University. The detailed processing parameters are summarized in Supplementary Table 1.

**PhLP2A-TRiC.** Movies were aligned in patches and CTF parameters were estimated in patches using cryoSPARC. Utilizing template-based autopicking in cryoSPARC v3.3, particles were initially picked and after 2D classification, they were used to train Topaz[55] and successfully picked 2,691,733 good particles. Among them, 2,285,466 particles were subjected to 3D heterogeneous refinement after ab initio model in cryoSPARC. Further 3D classification and CTF refinements were performed and non-uniform refinement was performed at last using 1,796,900 particles yielding a 3.05 Å consensus map of PhLP2A-TRiC based on the gold-standard Fourier shell correlation (FSC) at 0.143. For compositional analysis, heterogeneous refinement yielded one PhLP2A bound TRiC (486,149) and two PhLP2A bound TRiC (1,311,220). To further refine PhLP2A density, the consensus map was splitted into two half rings and merged to double the particle population. Then, the apical domain of TRiC and PhLP2A was masked and 3D classification was performed without alignment in RELION 4.0. CCT3 bound and CCT4 bound PhLP2A fraction were independently selected and locally refined in cryoSPARC again. As a result, CCT3 bound PhLP2A map and CCT4 bound PhLP2A map were obtained with 3.82 and 4.22 Å

resolution, respectively. This analysis workflow is illustrated in Supplementary Fig. 1b.

**Closed PhLP2A-TRiC with ATP/AlFx.** Movies were aligned in patches and CTF parameters were estimated with patch CTF correction in cryoSPARC. After the initial template-based picking of 833,250 particles, 282,298 particles were selected by 2D classification. After selection, ab initio and a few rounds of heterogeneous refinement resulted in 220,847 particles of closed TRiC (83.8%) and 42,742 particles of open TRiC (16.2%). Then, the inner chamber of closed TRiC density was masked followed by 3D classification without alignment in RELION. 30,911 particles (14.0%) contained one PhLP2A density and 21,028 particles (9.5%) showed two PhLP2A densities in the chamber. Since two PhLP2A densities exhibited the best density feature of PhLP2A, only TRiC containing two PhLP2A particles was pooled and imported to cryoSPARC. At last, after local motion correction and non-uniform refinement, 3.24 Å of closed PhLP2A-TRiC complex map was obtained. For the illustration purpose, we sharpened PhLP2A density independently from TRiC subunit densities as they showed varying resolution. PhLP2A maps were then segmented and shown in different thresholds for illustration. This analysis workflow is illustrated in Supplementary Fig. 5b.

**PFD-PhLP2A-TRiC.** Movies were aligned and CTF parameters were estimated in patches using cryoSPARC. 402,311 particles were picked using template matching and 2D classification was performed. Then, 232,059 particles were picked and ab initio reconstruction and multiple heterogeneous refinement was performed. 109,228 particles were used to reconstruct the consensus map of 3.84 Å. Since the noisy density around the PFD binding site and PhLP2A binding site was observed, further classification was performed. First, heterogeneous refinement giving PFD-TRiC maps in cryoSPARC as references successfully pooled 50,832 particles of TRiC with PFD density. The resolution was further pushed by discarding bad PFD-containing particles yielding 4.19 Å resolution of the PFD-TRiC map. Further masked 3D classification was performed but no PhLP2A density was detected. Meanwhile, a 3D reconstructed map without PFD was exported to RELION and 3D classification without alignment was performed near the PhLP2A binding site. 25,226 particles showed PhLP2A density while 25,129 particles were reconstructed as an apo-like structure. This analysis workflow is illustrated in Supplementary Fig. 4d.

**Truncated PhLP2A-TRiC.** (i) NTD-TXD of PhLP2A: Movies were aligned and CTF parameters were estimated in patches using cryoSPARC. 700,078 particles were picked using template matching and 2D classification was performed. Then, 588,578 particles were picked and ab initio reconstruction and multiple heterogeneous refinement was performed. 120,405 particles were used to reconstruct the consensus map of 4.11 Å. Since the noisy density around the NTD-TXD of the PhLP2A binding site was observed, further classification was performed. The apical domain of TRiC and PhLP2A was masked and 3D variability was performed in cryoSPARC. 23,847 particles (19.8%) contained NTD-TXD of PhLP2A density in the chamber. This analysis workflow is illustrated in Supplementary Fig. 7a. (ii) TXD of PhLP2A: Movies were aligned and CTF parameters were estimated in patches using cryoSPARC. 128,028 particles were picked using template matching and 2D classification was performed. Then, 33,767 particles were picked and ab initio reconstruction and multiple heterogeneous refinement was performed. 15,519 particles were used to reconstruct the consensus map of 7.24 Å. (iii) TXD-CTD of PhLP2A: Movies were aligned and CTF parameters were estimated in patches using cryoS-PARC. 125,947 particles were picked using template matching and 2D classification was performed. Then, 27,116 particles were picked and ab initio reconstruction and multiple heterogeneous refinement was

performed. 20,959 particles were used to reconstruct the consensus map of 3.78 Å.

**Truncated PhLP2A-TRiC with ATP/AlFx.** (i) NTD-TXD of PhLP2A: Movies were aligned and CTF parameters were estimated in patches using cryoSPARC. 899,293 particles were picked using template matching and 2D classification was performed. Then, 216,756 particles were picked and ab initio reconstruction and multiple heterogeneous refinement was performed. 157,313 particles were used to reconstruct the consensus map of 4.38 Å. Since the noisy density inside the TRiC chamber was observed, further classification was performed. The inner chamber of closed TRiC density was masked and 3D variability was performed in cryoSPARC. 12,976 particles (8.25%) contained NTD-TXD of PhLP2A density in the chamber. This analysis workflow is illustrated in Supplementary Fig. 7a. (ii) TXD of PhLP2A: Movies were aligned and CTF parameters were estimated in patches using cryoSPARC. 232,578 particles were picked using template matching and 2D classification was performed. Then, 57,473 particles were picked and ab initio reconstruction and multiple heterogeneous refinement was performed. 24,924 particles were used to reconstruct the consensus map of 6.61 Å. (iii) TXD-CTD of PhLP2A: Movies were aligned and CTF parameters were estimated in patches using cryoSPARC. 291,069 particles were picked using template matching and 2D classification was performed. Then, 106,204 particles were picked and ab initio reconstruction and multiple heterogeneous refinement was performed. 39,112 particles were used to reconstruct the consensus map of 4.60 Å.

**PhLP2A-β-actin-TRiC with ATP/AlFx.** Movies were aligned and CTF parameters were estimated in patches using cryoSPARC. 2,197,105 particles were picked using template-based picking followed by few rounds of 2D classification. 162,754 particles were subjected to 3D classification and open TRiC (14,924 particles) and closed TRiC 3D density maps (109,373 particles) were reconstructed. Closed TRiC was then further refined using CTF refinement and non-uniform refinement yielding consensus map of 3.38 Å resolution. Particles were targeted to 3D variability analysis using the inner chamber mask and classified 57.5% empty TRiC (62,899 particles) and 42.5% occupied TRiC (46,474 particles). Particles showing empty chambers were discarded and the rest of the particles were classified using 3D variability again to improve resolution followed by non-uniform refinement yielding 4.42 Å resolution of the structure (8,378 particles). This analysis workflow is illustrated in Supplementary Fig. 7b.

## Model building

**PhLP2A-TRiC.** The previous model of TRiC (PDB ID: 6NRA) was used as a reference to rigid-body fitting in the map. The structure of a thioredoxin-fold domain of human phosducin-like 2 (PDB code: 3EVI) and TXD-CTD of the AlphaFold-predicted model were used for the initial reference for the rigid-body fitting. After being fitted onto the density map, the initial model was manually refined in COOT[56] and further refined in Phenix real space refinement with default parameters[57,58].

**PFD-PhLP2A-TRiC.** The previously reported PFD-TRiC model (PDB:7W7U)[12] was used for the map model fitting and the illustration.

**PhLP2A-CCT3.** H3 of PhLP2A (residues 63–89) and CCT3 predicted using AlphaFold was used as an initial model and fitted onto the map. TXD of PhLP2A and CCT3 from PhLP2A:TRiC consensus model in the open state was used for fitting TXD of PhLP2A. After manual refinement of TXD of PhLP2A and CCT3 in COOT[56], the model was refined using MDFF and Phenix real space refinement with default options[57,58] for few rounds.

**PhLP2A-CCT4.** AlphaFold-predicted model of H3 of PhLP2A and CCT4 was used as an initial model and rigidly fitted onto the density map. TXD of PLP2A and CCT4 from the consensus model was used to fit the TXD of PhLP2A. The model was refined in COOT manually[56], and further adjusted using MDFF and Phenix real space refinement with default options[57,58].

**PhLP2A-TRiC with ATP/AlFx.** The model of TRiC in the closed form (PDB ID: 7LUM) and TXD of PhLP2A from AlphaFold prediction was rigidly fitted into the density map. H2 and H3 of Alphafold-predicted PhLP2A were used to fit the additional density extending from TXD of PhLP2A. The model was then manually adjusted in COOT[56] and further refined using MDFF and Phenix real space refinement with default parameters[57,58].

**PhLP2A-β-actin-TRiC with ATP/AlFx.** Previous model of the complex of PhLP2A-β-actin-TRiC in the closed form (PDB ID: 7NVM) and built model of PhLP2A-TRiC in the closed form from this study were fitted onto the map and used as initial models. The model of γ-actin in the initial model was exchanged with the model of β-actin from AlphaFold prediction. After rigid-body fitting, the models were manually adjusted in COOT. Further refinement was performed using MDFF and Phenix real space refinement with default parameters.

All models are validated by Phenix Comprehensive Validation (cryoEM)[59] and Q-score[60].

## AlphaFold prediction

ColabFold[32] prediction of version released on 2022/7/13 is used to predict the full-length model of PhLP2A. Default multisequence alignment pipeline and parameters are used without alteration. A predicted model deposited in AlphaFold DB (https://Alphafold.ebi.ac.uk/entry/Q9H2J4) was additionally presented. Complexes of CCT3-PhLP2A and CCT4-PhLP2A are predicted using the same version of ColabFold with default parameters and MSA pipeline[32] without any relaxation nor templates. Chain break between each component is specified with a colon to predict the heterodimeric complex.

## Sequence alignment and phylogenetic tree building

HHMER[61] was used to find hits of phosducin and phosducin-like protein variants from the eukaryotic system. Human phosducin and phosducin-like protein variants were used as a template for search and hits with low e-values were manually pooled. Also, sequences of cd02957 from NCBI which corresponds to the phosducin-like family were manually pooled. Total 71 sequences were used to generate multiple sequence alignments using t-coffee[62]. The phylogenetic tree was generated by FastTree[63] and visualized by Dendroscope[64]. The protein residue conservation score was calculated based on Shannon entropy scores with default parameters[65]. Then, based on the PhLP2A structure, aligned sequences were divided into 1–64 (H1-H2), 65–88 (H3), 89–199 (TXD), 200–239 (CTD) and the calculated scores of residues were averaged within each subgroup.

## Generation of residue conservation colored surface model

For analyzing residue conservation of CCT, PhLP2A, PFD, Consurf[33] was used with default parameters. Briefly, sequences of CCT1–8, PhLP2A, PFD subunits were used as input independently in Consurf to find hits and multiple sequence alignment (MSA) for each subunit was made with default settings in Consurf. Then, each residue conservation score was calculated based on MSA in Consurf. The surface models were colored and visualized in Chimera[66]. For PhLP1, PhLP3, the model was generated using homology modeling[67] and MSA for the conservation score calculation was generated using the grouped PhLP1 or PhLP3 sequences from the phylogenetic tree.

## Logo plot generation

Residue conservation was compared across 150 mainly eukaryotic with some archeal species for residues participating in the interaction with PhLP2A or PFD. Logo plots were generated using WebLogo 3[68].

## Reporting summary

Further information on research design is available in the Nature Portfolio Reporting Summary linked to this article.

## Data availability

The 3D cryoEM density maps have been deposited in the Electron Microscopy Data Bank under the accession code EMD-35284 (TRiC-PhLP2A open consensus), EMD-35199 (TRiC-PhLP2A open, CCT3 focused), EMD-35280 (TRiC-PhLP2A open, CCT4 focused), EMD-35122 (TRiC-PhLP2A-ATP/AlFx), EMD-35335 (TRiC-PhLP2A-actin-ATP/AlFx). Coordinates for the 3D cryoEM density maps have been deposited in the Protein Data Bank under the accession code PDB 8I9U, 8I6J, 8I9Q, 8I1U and 8IB8, respectively. PhLP2A-PFD-TRiC experiments maps (apo-like TRiC, PFD bound TRiC, PhLP2A bound TRiC) are deposited as additional maps in EMD-35284. Truncated PhLP2A-TRiC experiments maps in open or closed states are deposited as additional maps in EMD-35284 and EMD-35122, respectively. The mass spectrometry proteomics data have been deposited to the ProteomeXchange Consortium via the PRIDE partner repository with the dataset identifier PXD 040144. Source data are provided with this paper.

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

## Acknowledgements

This work has been supported by the National Institutes of Health grant R01GM074074 to JF, the Korean National Research Foundation (2019R1C1C1004598, 2020R1A5A1018081, 2021M3A9I4021220, 2019M3E5D6063871, 2020R1A6C101A183) and SUHF foundation to S.-H.R, and the Korean National Research Foundation (RS-2023-00245941) to H. Kim. The cryoEM data was collected and processed at The Center for Macromolecular and Cell Imaging (CMCI), Institute for Basic Science (IBS), Global Science experimental Data hub Center (GSDC) at Korea Institute of Science facilities. We thank Dr. Bum Han Ryu (Institute for Basic Science (IBS), Korea) for supporting cryoEM data collection. We also thank Dr. M. Steinegger and Dongwook Kim from Steinegger lab for discussions on evolutionary analysis. We thank members of CMCI for their discussions and suggestions. We thank members of the Frydman lab for useful discussions and suggestions.

## Author contributions

Conceptualization, J.F., D.G. and S.-H.R.; Investigation, J.P, H.K. D.G., S.L., K.O. and A.L.; Writing, J.F., D.G., J.P., H.K., S.L. and S.-H.R.; Funding Acquisition, J.F. and S.-H.R.

## Competing interests

The authors declare no competing interests.
