## [Peer Review File · Nature Communications]

A structural vista of phosducin-like PhLP2A-chaperonin TRiC cooperation during the ATP-driven folding cycleReviewers' Comments:

Reviewer #1:

Remarks to the Author:

Park et al., NATCOMS-23-11493 "A structural vista of phosducin-like PhLP2A-chaperonin TRiC cooperation during the ATP-driven folding cycle"

CCT associates with cochaperones to fold important eukaryotic proteins. In this manuscript, Park and co-workers have reconstituted different CCT complexes with PhLP2a and actin. The samples have been analysed and their structures with the chaperonin determined by cryo-EM. ATP hydrolysis is used by the chaperonin to drive the conformational changes leading to protein folding. The samples were isolated recombinantly and then reconstituted. The structural findings include the CCT- PhLP2a apo complex (3.1 Å global resolution and 3.82 and 4.22 for focused refinements), the mutually excluding interaction of PhLP2a and prefoldin (PFD), which rendered two structures (one with PFD and another with PhLP2a) and the structure of PhLP2a with CCT and ATP/AlFx (3.2 Å) and also including actin (4.42 Å).

The structures have been combined with XL-mass spec to propose a mechanism of folding including PhLP2a and how this co-factor may help CCT to expand the range of substrates of the chaperonin. The work contains a substantial amount of nice data, and the paper is well-written and follows a logical description of the structures combined with some of the biochemical experiments to support the main conclusions. However, the paper has some points that need to be addressed by the authors before is ready for publication. Especially the fact that the citations tend to ignore contributions from other authors in this field.

Major points

1.- One of my "philosophical" points with CCT is the extensive use of ATP/AlFx by different groups working on this topic. While the structures from the isolated sample are representative of the ATP cycle of CCT, it is questionable that the mixture of that sample with ATP→AlFx could yield representative functional states. This compound mimics the transition state of hydrolysis. Because of this treatment, the authors observe a large fraction of the particles in the double closed state, with all the ATPase sites occupied. Reissmann et al Cell Rep 2012 have published that the ATP binding sites have a different occupation, supporting the different binding occupancy of ATPγS found in the CCT-Tubulin crystal structure by Munoz et al NSMB 2011. How do they control that this is not an artefactual conformation?

2.- Previous studies have shown that the nucleotide occupancy of the CCT subunits is different (Munoz et al., NSMB 2011., Reissman et al Cell Rep. 2012, Zhang et al NSMB 2016 and many more...) corresponding with a different affinity for certain subunits. In addition, several groups have proposed working mechanisms based on the asymmetry observed in the open form compared to the closed complex (Yebeles et al TIBS 2011, Gestaut et al COSB 2019). None of these points is reflected or elaborated in the paper.

3.- The PFD bound CCT structure is not included in the table, please include the data if the map has been deposited, also the coordinates used in the figures.

4.- The initial sentence contains the wrong citation

The ring-shaped chaperonin TCP-1 ring complex (TRiC, also called CCT) is a ~1 MDa hetero-oligomeric complex that has a double-chamber architecture, with each octameric ring formed by subunits CCT1-8 1.

There are reports of the subunit composition of CCT since a long time, please use a more appropriate

reference to support that CCT is composed by 8 subunits
i.e A.L. Horwich, et al. Two families of chaperonin: physiology and mechanism
Annu. Rev. Cell Dev. Biol., 23 (2007)

5.- P9L304-305 The asymmetric mechanism of CCT has been previously supported by Munoz et al
NSMB 2011 and further reviewed by Yebenes, et al., TIBS2011. Please include both citations.

Reviewer #2:

Remarks to the Author:

Chaperonin TRiC/CCT is essential for cellular proteostasis. In the cell, TRiC cooperates with other cochaperones, including prefoldin (PFD) and phospho-tyrosine-like proteins (PhLPs) to facilitate substrate protein folding. PFD stabilizes substrate proteins in a dynamic unstructured conformation and delivers them to the TRiC chamber. It remains poorly defined how PhLPs engage TRiC throughout the ATP-driven conformation cycle.

In the present study, the authors studied the structural and mechanistic interplay between TRiC and PhLP2A throughout the ATP-driven chaperonin cycle using purified components to reconstitute actin folding in the presence of TRiC, PFD and PhLP. Combining Cryo-EM and biochemical assays, in this manuscript, the authors show:

1. The structure of PhLP2A with open TRiC in the absence of substrate
2. That PhLP2A precludes PFD binding to open TRiC.
3. The architecture of PhLP2A with closed TRiC in the absence of substrate with a considerable rearrangement of the binding sites compared to the open conformation.
4. The PhLP2 domain dependency for binding to TRiC in open and closed states
5. The structure of closed TRiC with folded actin and PhLP2A

Overall, the data is of good quality and generally support the author's conclusions. I recommend publication of this paper, once the following points have been addressed.

1. The native gel binding assays of PFD and TRiC upon addition of PhLP2A show release of PFD by PhLP2A already at time 0. Can the authors determine the relative K_d values for PFD and PhLP2A binding to TRiC? Can a similar native gel analysis be done to analyze the effect of PFD on PhLP2A binding?

2. Model Fig 7D: Authors propose actin stays in the septum when PhLP2A binds to TRiC. What is the evidence for this?

3. Model Fig 7D: Do the authors have evidence that actin is completely released from PFD when excluded by PhLP2A?

4. The commonalities and differences with a recently published structural study on yeast TRiC-Plp2-actin/tubulin (Han et al., Science Advances 9, eade1207) should be discussed. In this study, substrate and Plp2 appear to populate opposite rings of TRiC.

5. Does PhLP2 H1 interact with actin before the chamber closes?

Reviewer #3:

Remarks to the Author:

In this manuscript, Di Ianni et al describe the use of a cryoEM-based integrative structural approach to model the structure and dynamics of the ATP-driven cycle of TRiC-PFD-PhLP2A interaction. My expertise is in the area of structural mass spectrometry, and I am not able to comment significantly on

the cryo-EM data. This strategy of combining structural mass spectrometry with cryoEM is essential to study in modern structural biology and this study perfectly exemplifies the power of such an approach. Crosslinking MS experiments are well conducted, and the relevant data are correctly deposited in PRIDE repository. The combination of using heavy/light (H12/D12) DSS, as chemical crosslinker, and xQuest, for the data analysis, provides reliable structural restraints for deriving the structure of protein complexes. It is worth noting that some of the authors have extensive expertise in applying crosslinking MS on chaperonin TRiC/CCT.

The crosslinks observed by the authors not only orthogonally confirms the cryoEM data but also allows to extend their cryoEM-derived model. In fact, the crosslinks allow to locate and orient several protein regions not visualized by cryoEM in the complex.

Overall, this is a well performed study and I recommend its publication on Nature Communications.

Response to Reviewers Comments:

We thank all reviewers for their constructive comments. Overall, the reviewers are enthusiastic about the importance of the question we address and the impact and novelty of our findings. The revised manuscript includes a comprehensive response to the suggestions and requests posed by reviewers 1 and 2. We have carefully addressed each concern, incorporating new experiments and citations as suggested. We also conducted additional experiments to clarify the interplay between PhLP2A and PFD with respect to their interaction with TRiC. These new experiments have led to new and very interesting insights, now shown in the revised Fig 3. First, we measured the apparent K_d of PhLP2A for TRiC, and find it to be ~170 nM, i.e. similar to that of PFD-TRiC (~160 nM, Gestaut, Roh et al 2019)¹. Strikingly, despite the similar apparent K_d , our new biochemical analyses demonstrate that PhLP2A does indeed displace PFD from TRiC in a concentration-dependent manner, but interestingly, PFD does not displace PhLP2A. Thus, these new experiments support our original structural intuition but also suggest that these cochaperones may constitute a sequential pathway of TRiC interaction. We feel that the new experiments address a major suggestion from the reviewers, additional edits have corrected the minor concerns, strengthening the manuscript greatly and we thank them for their constructive comments. Our point-by-point response to the Reviewers' Comments is detailed below:

Response to Reviewer #1:

Reviewer #1: Park et al., NATCOMS-23-11493 "A structural vista of phosducin-like PhLP2A-chaperonin TRiC cooperation during the ATP-driven folding cycle"

CCT associates with cochaperones to fold important eukaryotic proteins. In this manuscript, Park and co-workers have reconstituted different CCT complexes with PhLP2a and actin. The samples have been analyzed and their structures with the chaperonin determined by cryo-EM. ATP hydrolysis is used by the chaperonin to drive the conformational changes leading to protein folding. The samples were isolated recombinantly and then reconstituted. The structural findings include the CCT- PhLP2a apo complex (3.1 Å global resolution and 3.82 and 4.22 for focused refinements), the mutually excluding interaction of PhLP2a and prefoldin (PFD), which rendered two structures (one with PFD and another with PhLP2a) and the structure of PhLP2a with CCT and ATP/AIFx (3.2 Å) and also including actin (4.42 Å).

The structures have been combined with XL-mass spec to propose a mechanism of folding including PhLP2a and how this co-factor may help CCT to expand the range of substrates of the chaperonin. The work contains a substantial amount of nice data, and the paper is well-written and follows a logical description of the structures combined with some of the biochemical experiments to support the main conclusions. However, the paper has some points that need to be addressed by the authors before is ready for publication. Especially the fact that the citations tend to ignore contributions from other authors in this field.

We thank the Reviewer for being enthusiastic about our study and for recognizing its significance and quality. We have revised the manuscript to address all the points raised in the review, as described below.

Major points

QI-1. One of my “philosophical” points with CCT is the extensive use of ATP-AIFx by different groups working on this topic. While the structures from the isolated sample are representative of the ATP cycle of CCT, it is questionable that the mixture of that sample with ATP-AIFx could yield representative functional states. This compound mimics the transition state of hydrolysis. Because of this treatment, the authors observe a large fraction of the particles in the double closed state, with all the ATPase sites occupied. Reissmann et al Cell Rep 2012 have published that the ATP binding sites have a different occupation, supporting the different binding occupancy of ATPgSe found in the CCT-Tubulin crystal structure by Munoz et al NSMB 2011. How do they control that this is not an artefactual conformation?

RI-1. We understand and value the reviewers' concern regarding the significance of the doubly closed state resulting from the wide use of ATP-AIFx, as we did in this study. For that matter, having no ATP or ADP present to study the open state is also unphysiological. In physiological conditions, Group II chaperonins use ATP cycling to alternate between the open state and the closed state that creates a confined inner chamber, necessary for substrate folding. As discussed below, both EM and solution approaches indicate that during ATP cycling the “double closed” state kinetically dominates the cycle, while asymmetric and open states, presumably containing both apo- and ADP-bound subunits, are transiently populated^{2,3}. We and others introduce ATP-AIFx to experimentally stabilize the closed non-cycling trigonal-bipyramidal conformation of the γ -phosphate during hydrolysis, just as we use nucleotide-free conditions to stabilize the open state, these conditions allow us to obtain a more homogeneous population of open and closed complexes for structural analyses². Using ATP-AIFx helps achieve a homogeneous closed conformation that increases particle numbers in the double-ring closed state with cochaperone and substrates. However, the similarity of this closed state with others obtained in the presence of hydrolysable ATP⁴ lends confidence to the underlying assumptions of our analyses using ATP-AIFx.

We also note that using ATP-AIFx is widely adopted and frequently cited in the literature⁵⁻⁷. Nonetheless, these doubly closed state is likely physiologically relevant based on recent studies. In their 2019 study, Jin et al.⁴ reported that yeast TRiC utilizes nucleotides in an inter-ring positively coordinated manner and showed under physiological ATP levels TRiC predominantly adopts the double ring-closed conformations. More recently, Zhao et al.⁸ demonstrated the presence of both-ring closed conformations of Type II chaperonin through stochastic binding and incomplete occupation of ATP-AIFx. Importantly, Douglas et al.⁹ found that non-cycling conditions using ATP with AIFx did not affect the rate and efficiency of substrate folding by a group II chaperonin compared to ATP conditions. In a recent series of studies, Kelly et al.¹⁰ and Gestaut et al.¹¹ tested the relationship of substrate folding in non-cycling conditions using ATP-AIFx. Liu et al.¹² independently used either ATP or ATP-AIFx conditions to initiate the substrate-TRiC folding cycle and observed the folded tubulin state bound in both-ring closed TRiC. Together, these findings suggest that using ATP-AIFx effectively represents the functional and structural folding state of the double-ring closed TRiC conformation. As we also successfully visualize folded actin with coordinated PhLP2A in this study, we consider that the TRiC double-ring closed conformation induced by ATP-AIFx represents a relevant conformation.

With regards to the distinct affinity of subunits for ATP, we agree with the reviewer that the cooperativity of cochaperones binding to TRiC under dynamic conditions of ATP cycling and their role in substrate folding is of great interest. We also agree that the ATP occupancy state in these “double closed” complexes should be thoroughly investigated. These are complicated questions outside the scope of this study, which aims to define the structural underpinnings of PhLP2A binding to the open and closed TRiC states. Considering the reviewer’s comment, we include relevant reference in the manuscript.

QI-2. Previous studies have shown that the nucleotide occupancy of the CCT subunits is different (Munoz et al., NSMB 2011., Reissman et al Cell Rep. 2012, Zhang et al NSMB 2016 and many more...) corresponding with a different affinity for certain subunits. In addition, several groups have proposed working mechanisms based on the asymmetry observed in the open form compared to the closed complex (Yebebenes et al TIBS 2011, Gestaut et al COSB 2019). None of these points is reflected or elaborated in the paper.

RI-2. We thank the reviewers for this insightful suggestion. In addition to the suggested reference, we add also a reference to Jin et al.⁴ which reported an ensemble of cryoEM structures of yeast TRiC at various nucleotide concentrations providing a view of the asymmetric TRiC conformational landscape from open to closed states. We agree with the reviewer’s point and now mention in the Discussion that it will be interesting to study the association of PFD and PhLPs under conditions of ATP cycling as well as in the presence of folding substrates. This should be an important topic for future research to reveal the asymmetric coordination of the TRiC complex. Considering the reviewer’s comment, we discussed this aspect with the suggested references and modified the text accordingly.

QI-3. The PFD bound CCT structure is not included in the table, please include the data if the map has been deposited, also the coordinates used in the figures.

RI-3. We thank the reviewer for alerting us to the lack of clarity in the data availability of our cryoEM maps and models. When we analyzed the PFD-TRiC electron density map, the maps we generated agreed well with the previous structure (PDB:7W7U)¹¹. Therefore, we took advantage of the PDB:7W7U model reported with higher resolution than our map for the structural analysis. We clarified this in the legend for Figure 3B. We also clarified more details for EM density and PDB data deposition used in our study in the supplementary table 1, and Method. For the maps from the PFD-PhLP2A-CCT cryoEM experiment, we deposited all three maps (PFD-CCT, PhLP2A-CCT, CCT) as additional maps in EMD-35284 and clarified in the manuscript with the Data availability section.

QI-4. The initial sentence contains the wrong citation

The ring-shaped chaperonin TCP-1 ring complex (TRiC, also called CCT) is a ~1 MDa A hetero-oligomeric complex that has a double-chamber architecture, with each octameric ring formed by subunits CCT1-8 1.

There are reports of the subunit composition of CCT since a long time, please use a more appropriate reference to support that CCT is composed by 8 subunits
i.e A.L. Horwich, et al. Two families of chaperonin: physiology and mechanism

Annu. Rev. Cell Dev. Biol., 23 (2007)

RI-4. We now include the original citation reporting the first proteomic analysis of mammalian TRiC subunits from 1992¹³ and its first characterization of multiple subunits in yeast from 1997¹⁴.

QI-5. P9L304-305 The asymmetric mechanism of CCT has been previously supported by Munoz et al NSMB 2011 and further reviewed by Yebenes, et al., TIBS2011. Please include both citations.

RI-5. We are thankful that the reviewer picked up this point. We accepted the reviewer's comments by including the suggested references in the main text.

Response to Reviewer #2:

Reviewer #2 : Chaperonin TRiC/CCT is essential for cellular proteostasis. In the cell, TRiC cooperates with other cochaperones, including prefoldin (PFD) and phosducin-like proteins (PhLPs) to facilitate substrate protein folding. PFD stabilizes substrate proteins in a dynamic unstructured conformation and delivers them to the TRiC chamber. It remains poorly defined how PhLPs engage TRiC throughout the ATP-driven conformation cycle.

In the present study, the authors studied the structural and mechanistic interplay between TRiC and PhLP2A throughout the ATP-driven chaperonin cycle using purified components to reconstitute actin folding in the presence of TRiC, PFD and PhLP. Combining Cryo-EM and biochemical assays, in this manuscript, the authors show:

1. The structure of PhLP2A with open TRiC in the absence of substrate
2. That PhLP2A precludes PFD binding to open TRiC.
3. The architecture of PhLP2A with closed TRiC in the absence of substrate with a considerable rearrangement of the binding sites compared to the open conformation.
4. The PhLP2 domain dependency for binding to TRiC in open and closed states
5. The structure of closed TRiC with folded actin and PhLP2A

Overall, the data is of good quality and generally support the author's conclusions. I recommend publication of this paper, once the following points have been addressed.

We thank the Reviewer for recognizing the novelty, significant impact and data quality of our study. We have revised the manuscript to address all the points raised in the review, as described below.

Major points

QII-1. The native gel binding assays of PFD and TRiC upon addition of PhLP2A show release of PFD by PhLP2A already at time 0. Can the authors determine the relative K_d values for PFD and PhLP2A binding to TRiC? Can a similar native gel analysis be done to analyze the effect of PFD on PhLP2A binding?

RII-1. This was an excellent suggestion that prompted a new series of experiments shown in the revised Figure 3. We previously determined the apparent K_d of PFD for open (apo) TRiC using a native gel-

based mobility shift to be $K_d = \sim 160 \text{ nM}$)¹. We now used the same approach to determine the apparent K_d of PhLP2A for open TRiC to be 170 nM; remarkably similar to that of PFD. This new experiment is now shown in **Fig. 3c**. Next, we carried out reciprocal competition experiments assessing the ability of either PFD or PhLP2A to displace the other cochaperone from a preformed complex with open TRiC. Briefly, fluorescently labeled cochaperones (shown here and in Gestaut, Roh et al 2019¹ to be functional), were bound to GFP-TRiC and their displacement from the TRiC complex by unlabeled cochaperone was determined using a Native gel mobility shift assay. Representative experiments and the quantification of three independent repeats are now shown in **Fig. 3d**. Strikingly, despite their similar overall apparent K_d values, we observe an unequal ability of PFD and PhLP2A to displace the other cochaperone. While PhLP2A displaced up to 70% of TRiC-bound PFD, similar concentrations of PFD resulted in virtually no displacement (< 10%) of TRiC-bound PhLP2A. We rationalize these observations based on the distinct mode of binding of these TRiC cochaperones. While PFD binds primarily to a single site in the apical domain of CCT4, PhLP2A binds in a multivalent manner, with binding sites in both the apical domains of CCT3/4 (which overlaps with that of PFD) but also within the equatorial domain regions of CCT3/6. These additional interactions potentially contribute to PhLP2A's enduring association with TRiC even when PFD is added in excess. Taken together with our structural data that PFD and PhLP2A both engage with TRiC through common binding interfaces, *our study now establishes a compelling hypothesis for the interplay between these cochaperones in their interaction with TRiC, which is now summarized in the new Fig. 7d*. Basically, we propose that PFD delivers the substrate to TRiC and is subsequently displaced by PhLP2A, which then facilitates folding after closure following ATP hydrolysis. Of course the nature of these interactions in the cell should be critically tested *in vivo* in future studies.

QII-2. Model Fig 7D: Authors propose actin stays in the septum when PhLP2A binds to TRiC. What is the evidence for this?

RII-2. We thank the reviewer for pointing out this speculation in our illustration. We initially described the location of actin in open TRiC conformation based on the work of the Cambridge group¹⁰ and by analogy with the location of tubulin in Gestaut et al¹¹. These studies observed the substrate-induced density in the septum of TRiC, the first following IP of TRiC from cells, the second following delivery of tubulin from a PFD complex. In these studies, the substrate-induced density could be due to tethering of the substrate rather as cryoEM does not examine where the substrate binds, but rather where its largest density is located. Indeed, Hydrogen Deuterium Exchange (HDX) experiments on actin bound to TRiC indicate that non-native actin binds to the apical domains of TRiC¹⁵. More recently, Han et al¹⁶ observed extra-density in the chamber close to apical and equatorial domains of open TRiC. Collectively, these studies suggest that non-native actin is dynamically localized to multiple regions within the open TRiC chamber. We have modified the panel in Fig. 7d to better reflect the existing data and modified the text accordingly.

QII-3. Model Fig 7D: Do the authors have evidence that actin is completely released from PFD when excluded by PhLP2A?

RII-3. The reviewer's observation is indeed valid since we do not include substrate in our competition experiments and neither have other groups reconstituted the entire folding pathway. In fact, Balchin et al¹⁵ observe efficient folding of actin in the absence of either PFD or PhLPs, and a previous study by

Gestaut et al.¹, demonstrated that PFD enhances the kinetics and processivity of TRiC-mediated actin folding in the absence of PhLP2A. Since we don't provide any further evidence for the effect of PhLP2A during PFD-mediated substrate delivery in this study we have modified fig. 7d to *remove the substrate* from the cochaperone interactions with open TRiC and have limited our description of the role of PhLP2A in the closed state to what is suggested by our structural analysis.

Q4.-The commonalities and differences with a recently published structural study on yeast TRiC-Plp2-actin/tubulin (Han et al., Science Advances 9, eade1207) should be discussed. In this study, substrate and Plp2 appear to populate opposite rings of TRiC.

We sincerely appreciate the reviewer's comment on the recently published reference¹⁶, which appeared during our review process. Han et al also provide valuable insights that expand the understanding of the yeast TRiC-Plp2 interaction. There are several commonalities as well as differences between the two studies, now discussed in the manuscript and here.

Structural comparison: Han et al, conducted an investigation on TRiC-Plp2 from yeast, which exhibits homology to the human TRiC complex with PhLP. While we did not explore the ternary complex of actin in open TRiC-PhLP2A conformation, Han et al., resolved a ternary complex structure of TRiC-Plp2-actin in nucleotide-free and ATP-bound open conformations. In the snapshot structure of closed TRiC Han et al observe folded actin in one chamber, and Plp2 in the opposite chamber with an overall architecture and specific CCT interaction that are almost identical to what we identify in the human system in this study. However, unlike our study, they do not find the extended helices of Plp2 making direct contact with actin. It is noteworthy that the open conformation of the TRiC-PLP2 complex in Han et al, shows one Plp2 molecule in the cis-TRiC chamber and additional densities present in the trans-TRiC chamber that Han et al suggest representing substrate bound to specific CCT subunits. Han et al. also discusses significant density in the septum as a tangle of CCT tails, although we propose this septum density may also contain substrate tethered by CCT tails, as supported by their own cross-linking mass analysis. Since Han et al does not demonstrate the role of PFD or Plp2 in actin folding biochemically, any mechanistic inference of the structures should be considered with caution.

Differences in substrate-PhLP substrate positioning: It is noteworthy that Han et al. utilized chemically denatured actin to deliver the substrate throughout their study, while our approach involves using copurifying actin-PFD to introduce unstructured actin into the TRiC system. In a separate study by Gestaut et al.¹, it was shown that substrates bind to PFD in an unstructured state, which can undergo folding upon delivery to TRiC in the presence of ATP. This discrepancy in experimental design may influence the determination of the substrate binding location within the open TRiC chamber. For tubulin, we observe PFD delivers the substrate deep inside the TRiC chamber as compatible with the density in the septum that was attributed as substrate in the tubulin study¹. Consequently, the mechanism of PFD-mediated actin delivery to open TRiC remains uncertain and necessitates further investigation. Addressing this aspect in future studies is essential to shed light on the intricacies of chaperone-assisted protein folding and the specific recognition of substrates.

Additional unique contributions from our study: Our study makes a number of important and unique contributions that provide a deeper understanding of the structural and functional aspects of PhLP-TRiC interactions. **(1)** we present a detailed analysis of the binding and structural interactions of the PhLP2A

domains to TRiC in the open and closed states, which provide novel insights into the complex movements undergone by the cochaperone upon ATP hydrolysis by TRiC. **(2)** we place the structural understanding of our study in an evolutionary perspective of the PhLP family. The combination illuminates the possible functions and TRiC regulation of the entire cochaperone family. **(3)** we examine the interactions of PFD and PhLP2A with TRiC, which are now illuminated by the new set of experiments suggested by the reviewer and provide insight into the organization of the TRiC cochaperone networks.

Q5.-Does PhLP2 H1 interact with actin before the chamber closes?

The question regarding the precise timing of the interaction between PhLP2A H1 and actin is excellent and warrants further investigation. Currently, we lack evidence to pinpoint the exact moment when the PhLP2A H1 interacts with actin so we can only speculate about these events. It is worth noting that previous HDX experiments show the N-terminus of Subdomain 1 of actin¹⁵, which is in close proximity to the PhLP2A H1 binding site, folds relatively late in the closed TRiC chamber. We observe that the reorientation of PhLP2A N-terminal helices brings them closer to actin as TRiC undergoes ATP-driven closing. We thus speculate that formation of the PhLP2A H1 binding determinant during actin folding precedes H1 binding, but it is also possible that H1 templates these events. Absent further direct data on the mechanism of substrate folding, we prefer to limit the speculation and defer to a future analysis of the reaction.

Response to Reviewer #3:

Reviewer #3 : In this manuscript, Di Ianni et al describe the use of a cryoEM-based integrative structural approach to model the structure and dynamics of the ATP-driven cycle of TRiC-PFD-PhLP2A interaction. My expertise is in the area of structural mass spectrometry, and I am not able to comment significantly on the cryo-EM data. This strategy of combining structural mass spectrometry with cryoEM is essential to study in modern structural biology and this study perfectly exemplifies the power of such an approach.

Crosslinking MS experiments are well conducted, and the relevant data are correctly deposited in PRIDE repository. The combination of using heavy/light (H12/D12) DSS, as chemical crosslinker, and xQuest, for the data analysis, provides reliable structural restraints for deriving the structure of protein complexes. It is worth noting that some of the authors have extensive expertise in applying crosslinking MS on chaperonin TRiC/CCT.

The crosslinks observed by the authors not only orthogonally confirms the cryoEM data but also allows to extend their cryoEM-derived model. In fact, the crosslinks allow to locate and orient several protein regions not visualized by cryoEM in the complex.

Overall, this is a well performed study and I recommend its publication on Nature Communications.

We thank the Reviewer for being enthusiastic about our study and for recognizing its significance and quality.

References Cited for the Responses to the Reviewers

1. Gestaut, D. et al. The Chaperonin TRiC/CCT Associates with Prefoldin through a Conserved Electrostatic Interface Essential for Cellular Proteostasis. *Cell* **177**, 751-765 e15 (2019).
2. Meyer, A.S. et al. Closing the folding chamber of the eukaryotic chaperonin requires the transition state of ATP hydrolysis. *Cell* **113**, 369-81 (2003).
3. Booth, C.R. et al. Mechanism of lid closure in the eukaryotic chaperonin TRiC/CCT. *Nat Struct Mol Biol* **15**, 746-53 (2008).
4. Jin, M. et al. An ensemble of cryo-EM structures of TRiC reveal its conformational landscape and subunit specificity. *Proc Natl Acad Sci U S A* **116**, 19513-19522 (2019).
5. Fisher, A.J. et al. X-ray structures of the myosin motor domain of Dictyostelium discoideum complexed with MgADP.BeFx and MgADP.AIF4. *Biochemistry* **34**, 8960-72 (1995).
6. Rayment, I., Smith, C. & Yount, R.G. The active site of myosin. *Annu Rev Physiol* **58**, 671-702 (1996).
7. Sprang, S.R. G protein mechanisms: insights from structural analysis. *Annu Rev Biochem* **66**, 639-78 (1997).
8. Zhao, Y., Schmid, M.F., Frydman, J. & Chiu, W. CryoEM reveals the stochastic nature of individual ATP binding events in a group II chaperonin. *Nat Commun* **12**, 4754 (2021).
9. Douglas, N.R. et al. Dual action of ATP hydrolysis couples lid closure to substrate release into the group II chaperonin chamber. *Cell* **144**, 240-52 (2011).
10. Kelly, J.J. et al. Snapshots of actin and tubulin folding inside the TRiC chaperonin. *Nat Struct Mol Biol* **29**, 420-429 (2022).
11. Gestaut, D. et al. Structural visualization of the tubulin folding pathway directed by human chaperonin TRiC/CCT. *Cell* **185**, 4770-4787 e20 (2022).
12. Liu, C. et al. Pathway and mechanism of tubulin folding mediated by TRiC/CCT along its ATPase cycle revealed using cryo-EM. *Commun Biol* **6**, 531 (2023).
13. Frydman, J. et al. Function in protein folding of TRiC, a cytosolic ring complex containing TCP-1 and structurally related subunits. *EMBO J* **11**, 4767-78 (1992).
14. Lin, P. & Sherman, F. The unique hetero-oligomeric nature of the subunits in the catalytic cooperativity of the yeast Cct chaperonin complex. *Proc Natl Acad Sci U S A* **94**, 10780-5 (1997).
15. Balchin, D., Milicic, G., Strauss, M., Hayer-Hartl, M. & Hartl, F.U. Pathway of Actin Folding Directed by the Eukaryotic Chaperonin TRiC. *Cell* **174**, 1507-1521 e16 (2018).
16. Han, W. et al. Structural basis of plp2-mediated cytoskeletal protein folding by TRiC/CCT. *Sci Adv* **9**, eade1207 (2023).

Reviewers' Comments:

Reviewer #1:

Remarks to the Author:

The answers to my questions are unsatisfactory, especially the first one.

RI-1. We understand and value the reviewers' concern regarding the significance of the doubly closed state resulting from the wide use of ATP-AIFx, as we did in this study. For that matter, having no ATP or ADP present to study the open state is also unphysiological.

I am sorry but this is an evasive response to a point that is important. Two negatives do not make a positive.

Using ATP-AIFx helps achieve a homogeneous closed conformation that increases particle numbers in the double-ring closed state with cochaperone and substrates.

This is the real reason; that homogeneous populations are better at obtaining a high-resolution structure by cryo-EM. The point is whether this homogeneous population is representative or not within the CCT folding cycle.

We also note that using ATP-AIFx is widely adopted and frequently cited in the literature⁵⁻⁷.

Does not mean it is correct...In my opinion, this is an assumption not justified which has been maintained due to processing purposes and the lack of methods to deal with the heterogeneous populations by cryo-EM. However, this is not the case now.

With regards to the distinct affinity of subunits for ATP, we agree with the reviewer that the cooperativity of cochaperones binding to TRiC under dynamic conditions of ATP cycling and their role in substrate folding is of great interest. We also agree that the ATP occupancy state in these "double closed" complexes should be thoroughly investigated. These are complicated questions outside the scope of this study, which aims to define the structural underpinnings of PhLP2A binding to the open and closed TRiC states. Considering the reviewer's comment, we include relevant reference in the manuscript

Finally, the authors do not address this question. They considered it important but not the aim of this manuscript. I think is contradictory.

I think the authors need to rework these answers.

Reviewer #2:

Remarks to the Author:

The authors have addressed all my comments either with new experiments or editorial changes. I have no further questions and recommend publication.

Point-by-point response to the reviewers' comments

Original comment.

One of my "philosophical" points with CCT is the extensive use of ATP-AIFx by different groups working on this topic. While the structures from the isolated sample are representative of the ATP cycle of CCT, it is questionable that the mixture of that sample with ATP-AIFx could yield representative functional states. This compound mimics the transition state of hydrolysis. Because of this treatment, the authors observe a large fraction of the particles in the double closed state, with all the ATPase sites occupied. Reissmann et al Cell Rep 2012 have published that the ATP binding sites have a different occupation, supporting the different binding occupancy of ATPgSe found in the CCT-Tubulin crystal structure by Munoz et al NSMB 2011. How do they control that this is not an artefactual conformation?

Second comment.

"We understand and value the reviewers' concern regarding the significance of the doubly closed state resulting from the wide use of ATP-AIFx, as we did in this study. For that matter, having no ATP or ADP present to study the open state is also unphysiological."

I am sorry but this is an evasive response to a point that is important. Two negatives do not make a positive.

"Using ATP-AIFx helps achieve a homogeneous closed conformation that increases particle numbers in the double-ring closed state with cochaperone and substrates."

This is the real reason; that homogeneous populations are better at obtaining a high-resolution structure by cryo-EM. The point is whether this homogeneous population is representative or not within the CCT folding cycle.

"We also note that using ATP-AIFx is widely adopted and frequently cited in the literature⁵⁻⁷."

Does not mean it is correct...In my opinion, this is an assumption not justified which has been maintained due to processing purposes and the lack of methods to deal with the heterogeneous populations by cryo-EM. However, this is not the case now.

"With regards to the distinct affinity of subunits for ATP, we agree with the reviewer that the cooperativity of cochaperones binding to TRiC under dynamic conditions of ATP cycling and their role in substrate folding is of great interest. We also agree that the ATP occupancy state in these "double closed" complexes should be thoroughly investigated. These are complicated questions outside the scope of this study, which aims to define the structural underpinnings of PhLP2A binding to the open and closed TRiC states. Considering the reviewer's comment, we include relevant reference in the manuscript"

Finally, the authors do not address this question. They considered it important but not the aim of this manuscript. I think is contradictory. I think the authors need to rework these answers.

Response.

We had misconstrued certain aspects of the reviewer's initial comment. Thank you for providing clarification. We value the reviewer's inquiry concerning the representativeness of the double-ring closed TRiC structure induced by ATP-AIFx in our study. We conducted a thorough examination of existing literature on TRiC's conformational changes under various ATP conditions. It should be noted that studies reporting differences in nucleotide occupancy within the ATP binding pockets of CCTs, such as those by Munoz et al. (NSMB 2011)¹ and Jin et al. (PNAS 2019)², have exclusively observed in the open conformation of TRiC. In contrast, all closed TRiC structures in both ATP and ATP/AIFx consistently exhibit full nucleotide occupancy. Specifically, we referred to Jin et al.'s detailed structural analyses of yeast TRiC, which revealed a close connection between TRiC's conformations and the ATP cycle. Their work notably reported that when TRiC encounters 0.2mM ATP, a significant portion of TRiC can adopt a double-ring closed state. Importantly, this ATP-induced double-ring structure showed

nucleotides bound in all ATP binding pockets and closely resembled the structure generated in the presence of ATP+AIFx during their compatible experiment. Furthermore, it's noteworthy to mention that Reissmann et al (Cell Rep 2012)³ investigated the asymmetric binding affinity of nucleotides using both ATP and ATP/AIFx, yielding compatible results in both conditions and confirmed the use of AIFx doesn't significantly change the occupancy of bound nucleotides on each CCT subunit. These references suggest that full nucleotide occupancy in the closed TRiC is not an artifact caused by AIFx and varied nucleotide binding may be influenced by TRiC's different structural conformations.

Additionally, recent research by Liu et al. in 2023⁴ provided further evidence regarding the structures of human TRiC in the presence of both ATP and ATP+AIFx. 27% of TRiC particles exhibited a double-ring closed state when stimulated by ATP, mirroring the structure seen with ATP+AIFx, where 60% of the population displayed this conformation. In particular, Liu et al., confirmed their structure with ATP condition showing that the double-ring closed state induced by ATP has complete occupancy in the nucleotide-binding pocket and still captures the substrate in the same manner with ATP+AIFx. Consistently, Douglas et al.⁵ showed that the condition of ATP with AIFx still efficient for TRiC to fold substrate compared to ATP condition. Collectively, these meticulously controlled previous experiments support that the TRiC in ATP+AIFx can fold substrate and mimic one of the TRiC conformational landscapes driven by hydrolysis of natural ATP.

We then conducted a detailed examination of nucleotide binding patterns within our structures. This analysis encompassed the quantification of nucleotide binding based on the relative density; a method designed to eliminate the risk of arbitrary interpretations that might be influenced by differences in contour levels utilized in the maps. Inside the open TRiC structure, we pinpointed regions of density that were indicative of ADP binding within the ATP binding pocket (Figure R1). We did observe the pattern of partial nucleotide occupancy with a more pronounced ADP signal discerned in the lower affinity hemisphere of TRiC, specifically in CCT6, 8 and 3, as opposed to the higher affinity side of TRiC like CCT2, 5 and 7 (Figure R1). This observation consistently reaffirms that purified open TRiC exhibits a certain level of endogenous ADP in the lower affinity hemisphere. This mode of nucleotide binding aligns closely with the findings from earlier investigations of open TRiC structures.

Figure R1

Figure R1. **a** Top) cryo-EM structure of open TRiC-PhLP2A complex without nucleotide supplemented from this study, bottom) Density of ATP binding pocket occupied with nucleotide on cis- and trans-ring of TRiC complex. Density corresponds to the nucleotide on each ATP binding pocket is segmented and colored as red. **b** Zoom-in view on ATP binding pocket on each CCT subunit, showing different volume of density depending on different subunits. While CCT8/6/3 show density where nucleotide can fit on, other CCT subunits, especially CCT5, show relatively weak densities on the pocket. **c** A graph of relative intensity of ATP binding pocket on each CCT subunit. Mass of each segmented volume was calculated as a ratio, with the maximum value serving as the denominator. CCT3/6/8 show significantly higher intensity in comparison with others. Deep blue and sky blue: relative mass of each CCT subunit in cis-ring and trans-ring, respectively. Light gray line: average of the ratios.

Importantly, when examining the ATP+AIFx-induced closed structures, we noted that the overall architecture of our TRiC closely resembled that of the natural ATP-induced closed structure (PDB 7X6Q)⁴ (Figure R2a). All subunits exhibited a backbone RMSD of less than 2 angstroms when compared to the native ATP-induced configuration (Figure R2b). Furthermore, our structure displayed full nucleotide occupancy in a manner similar to what is observed in ATP-induced structures (Figure R2c). In a quantitative analysis, we observed that nucleotide intensities were relatively consistent across subunits, which is also consistent with ATP-induced structures (Figure R2c). As a result, we conclude that the ATP+AIFx analog in this study does not cause an artifactual conformation and effectively captures the complex in a representative closed state with full nucleotide occupancy, replicating the post-hydrolysis ATP state.

Figure R2

Figure R2. **a** Two cryo-EM structures of TRiC in closed state, induced by either ATP or ATP/AIFx. **b** Superimposition of each CCT subunit in the two structures, which all possess overall similar conformation. **c** Zoom-in view on ATP binding pocket of each CCT subunit on left) ATP-induced closed TRiC or right) ATP/AIFx-induced closed TRiC. In both case, ATP binding pockets on all the CCT subunits show significant densities corresponding to the nucleotides reside in. **d** Segmented density of ATP binding pockets on cis- and trans-ring of each structure of closed TRiC, which is red-color labelled. **e** Graphs of relative intensity of ATP binding pockets of ATP-treated or ATP/AIFx treated closed TRiC, adopting same quantitation and calculation as in Figure R1 c. In both structures, deviation between volumes of ATP binding pocket in each CCT subunit was slight comparing to those in open TRiC. Graphs are color coded as same as the previous graph.

To recap, considering the insights from prior references and our structural analysis, we find it reasonable to conclude that AIFx does not introduce artifacts in the TRiC structure but rather accurately portrays the structure in the post-hydrolysis state during TRiC's ATP hydrolysis cycle. As mentioned by the reviewer, we did, in fact, utilize ATP+AIFx to facilitate a high-resolution structural analysis by capturing a homogeneous population. An added benefit of employing AIFx is its capacity to induce one-way conformational changes, thus preventing the reopening and recycling of TRiC. By adopting this approach with AIFx, we not only achieved a high-resolution structural representation but also obtained a well-defined chemical state simplifying the interpretation of the meaning of the enclosed cochaperone and substrate. The reviewer might contemplate conducting the cryoEM experiment solely in the presence of native ATP and then dissecting different nucleotide binding and multiple diverse structures through computational analysis. Despite the significant advancements in addressing heterogeneity via cryoEM techniques, it remains a formidable challenge to provide a comprehensive characterization of the inherently diverse and heterogeneous features of pseudo-symmetric molecules like TRiC, especially when dealing with relatively faint and ensemble signals, such as those associated with ligand binding. It pertains to a comprehensive structural and biochemical characterization aimed at elucidating how different CCT subunits exhibit distinct nucleotide binding, exchange, and hydrolysis rates, and how these asymmetric dynamics mechanistically drive transitions in TRiC's structural conformations. This characterization is of utmost importance for understanding TRiC itself and will necessitate extensive and multi-modal approaches encompassing both structural and biochemical analyses in the future. As we focus on the initial characterization of TRiC-cochaperone interplay in this manuscript, we consider TRiC's asymmetric nucleotide relationship to be handled as an independent story.

We are in agreement with the reviewer's recommendation and have implemented changes to the Figures. These updates encompass panels displaying ATP binding pockets for both open and closed TRiC structures. We have also adjusted the text to provide ideas of nucleotide binding and a more comprehensive rationale with the use of AIFx.

Manuscript

The architecture of PhLP2A complex with open apo-state TRiC

The complex between open human TRiC and PhLP2A was generated using purified components and subjected to cryoEM to investigate its structure. We obtained a 3.08 Å resolution consensus map (Fig. 1a, Supplementary Fig. 1a-d) containing PhLP2A-dependent density within the TRiC chamber, positioned close to the equatorial region. *The overall TRiC architecture closely mirrors previously established structural characteristics^{6,7}, maintaining robust equatorial domains and dynamic apical regions. Additionally, we noted a consistent pattern of partial nucleotide occupancy, with a more prominent nucleotide density observed in the lower affinity hemisphere of TRiC3, particularly in CCT6, CCT8, and CCT3, in contrast to the higher affinity side of TRiC, represented by CCT2, CCT5, and CCT7 (Supplementary Fig. 2).* Subsequent 3D classification and refinement revealed high-resolution features for this extra density (Supplementary Fig. 1b). We then fitted the crystal structure from the thioredoxin-fold domain of human PhLP2B to the density (PDB code: 3EVI)⁸ and the model aligned well with many bulky side chains corresponding to the cryoEM density. This provides strong evidence that the extra density represents PhLP2A (Fig. 1b).

The architecture of PhLP2A complex with ATP-AIFx induced closed state TRiC

TRiC conformation changes upon ATP hydrolysis, which induces lid closure over the central chamber⁹⁻¹¹. Since lid closure occludes the PFD binding site in CCT3/4 and induces the release of bound PFD from TRiC, we examined if PhLP2A forms a complex with closed TRiC. *In a prior study, Jin et al. demonstrated that TRiC maintains an identical structure with full nucleotide occupancy in the ATP binding pockets under both ATP and ATP/AIFx conditions. Since AIFx captures TRiC complex in a stable post-hydrolyzed state during TRiC's ATP hydrolysis cycle^{4,12-14}, we chose to employ the nucleotide analog ATP/AIFx to attain high-resolution structures.* We incubated the TRiC: PhLP2A in a 1:4 molar ratio with ATP/AIFx, which generates a stably closed TRiC conformation and subjected it to cryoEM analysis (Fig. 4a-(i)). The consensus map of closed TRiC reached 2.95 Å resolution with significant additional density inside the chamber. *We validated that our closed TRiC structure resembled that of the native ATP-induced structure including full nucleotide occupancy in a manner similar to what is observed in ATP-induced structures^{2,7} (Supplementary Fig 6).* Subsequently, focused classification on the inner TRiC chambers identified subpopulations of PhLP2A in either one (14.0%) or both chambers (9.5%) (Supplementary Fig. 4a-d). The TRiC subpopulation with PhLP2A in both chambers was used

to further refine the final map of the PhLP2A protein fully encapsulated inside the closed TRiC chamber at 3.24 Å resolution (Fig. 4a-(ii, iii)). Overall, side chain details for TRiC were clearly revealed on the map allowing us to build an atomic model. For PhLP2A, we could build a refined model spanning residues 27-214 (Fig. 4b-(i), Supplementary Fig. 4e). PhLP2A H1-2 (aa1-26) in the NTD and the CTD (aa 215-239) are not well resolved but have attributable densities that can be observed at a lower contour providing an approximate location of the NTD in proximity to CCT1 and CCT3 and of the CTD in proximity to the intermediate domains between CCT1 and CCT4 (Fig. 4b-(ii)).

Supplementary figures.

Supplementary Fig. 2 Analyses on ATP binding pocket in an open TRiC-PhLP2A complex **a** Top) cryo-EM structure of open TRiC-PhLP2A complex without nucleotide supplemented from this study, bottom) Density of ATP binding pocket occupied with nucleotide on cis- and trans-ring of TRiC complex. Density corresponds to the nucleotide on each ATP binding pocket is segmented and colored as red. **b** Zoom-in view on ATP binding pocket on each CCT subunit, showing different masses depending on different subunits. While CCT8/6/3 show density where nucleotide can fit on, other CCT subunits, especially CCT5, show relatively weak densities on the pocket. **c** A graph of relative intensity of ATP binding pocket on each CCT subunit. Mass of each segmented volume was calculated using ChimeraX Measure volume tool, and the ratio to the maximum volume of ATP binding pocket was presented. CCT3/6/8 shows significantly higher intensity in comparison with others. Deep blue and sky blue: mass of each CCT subunit in cis-ring and trans-ring, respectively. Light gray line: average of the ratio values.

Supplementary Fig. 6 Analyses on ATP binding pocket in a closed TRiC-PhLP2A complex **a** Top) cryo-EM structure

of closed TRiC-PhLP2A induced by 1mM ATP/AlFx. bottom) Density of ATP binding pocket occupied with nucleotide on cis- and trans-ring of TRiC complex. **b** Zoom-in view on ATP binding pocket on each CCT subunit. **c** A graph of relative intensity of ATP binding pocket on each CCT subunit. Mass of each segmented volume and the ratio to the maximum value from the molecule is calculated and presented as in Supplementary Fig. 2 c).

References

1. Munoz, I.G. et al. Crystal structure of the open conformation of the mammalian chaperonin CCT in complex with tubulin. *Nat Struct Mol Biol* **18**, 14-9 (2011).
2. Jin, M. et al. An ensemble of cryo-EM structures of TRiC reveal its conformational landscape and subunit specificity. *Proc Natl Acad Sci U S A* **116**, 19513-19522 (2019).
3. Reissmann, S. et al. A gradient of ATP affinities generates an asymmetric power stroke driving the chaperonin TRiC/CCT folding cycle. *Cell Rep* **2**, 866-77 (2012).
4. Liu, C. et al. Pathway and mechanism of tubulin folding mediated by TRiC/CCT along its ATPase cycle revealed using cryo-EM. *Commun Biol* **6**, 531 (2023).
5. Jiang, Y. et al. Sensing cooperativity in ATP hydrolysis for single multisubunit enzymes in solution. *Proc Natl Acad Sci U S A* **108**, 16962-7 (2011).
6. Gestaut, D. et al. The Chaperonin TRiC/CCT Associates with Prefoldin through a Conserved Electrostatic Interface Essential for Cellular Proteostasis. *Cell* **177**, 751-765 e15 (2019).
7. Han, W. et al. Structural basis of plp2-mediated cytoskeletal protein folding by TRiC/CCT. *Sci Adv* **9**, eade1207 (2023).
8. Lou, X., Bao, R., Zhou, C.Z. & Chen, Y. Structure of the thioredoxin-fold domain of human phosphoglucomutase-like protein 2. *Acta Crystallogr Sect F Struct Biol Cryst Commun* **65**, 67-70 (2009).
9. Cong, Y. et al. Symmetry-free cryo-EM structures of the chaperonin TRiC along its ATPase-driven conformational cycle. *EMBO J* **31**, 720-30 (2012).
10. Reissmann, S., Parnot, C., Booth, C.R., Chiu, W. & Frydman, J. Essential function of the built-in lid in the allosteric regulation of eukaryotic and archaeal chaperonins. *Nat Struct Mol Biol* **14**, 432-40 (2007).
11. Gestaut, D., Limatola, A., Joachimiak, L. & Frydman, J. The ATP-powered gymnastics of TRiC/CCT: an asymmetric protein folding machine with a symmetric origin story. *Curr Opin Struct Biol* **55**, 50-58 (2019).
12. Douglas, N.R. et al. Dual action of ATP hydrolysis couples lid closure to substrate release into the group II chaperonin chamber. *Cell* **144**, 240-52 (2011).
13. Kelly, J.J. et al. Snapshots of actin and tubulin folding inside the TRiC chaperonin. *Nat Struct Mol Biol* **29**, 420-429 (2022).
14. Gestaut, D. et al. Structural visualization of the tubulin folding pathway directed by human chaperonin TRiC/CCT. *Cell* **185**, 4770-4787 e20 (2022).

Reviewers' Comments:

Reviewer #1:

Remarks to the Author:

Although I do not fully agree with some of the arguments the authors have made a good effort to address the questions. I think the paper is ready for publication